# Molecular Targeting of the Human Epidermal Growth Factor Receptor-2 (HER2) Genes across Various Cancers

**DOI:** 10.3390/ijms25021064

**Published:** 2024-01-15

**Authors:** Elizabeth Rubin, Khine S. Shan, Shivani Dalal, Dieu Uyen Dao Vu, Adriana M. Milillo-Naraine, Delia Guaqueta, Alejandra Ergle

**Affiliations:** Memorial Cancer Institute, Pembroke Pines, FL 33028, USA; kshan@mhs.net (K.S.S.); sdalal@mhs.net (S.D.); dieuuyenvu@utexas.edu (D.U.D.V.); amilillonaraine@mhs.net (A.M.M.-N.); dguaqueta@mhs.net (D.G.); aergle@mhs.net (A.E.)

**Keywords:** HER2, HER2 discovery, HER2 role in cancers, HER 2 low, HER2DX, prognostic, predictive, biomarker, anti-HER therapies, HER2 landmark trials

## Abstract

Human epidermal growth factor receptor 2 (HER2) belongs to the ErbB family, a group of four transmembrane glycoproteins with tyrosine kinase activity, all structurally related to epidermal growth factor receptor (EGFR). These tyrosine kinases are involved in the transmission of cellular signals controlling normal cell growth and differentiation. If this transmission goes awry, it can lead to dysregulated growth of the cell. HER2 specifically can be implicated in the pathogenesis of at least eight malignancies. HER2 positivity quickly became a well-characterized indicator of aggressiveness and poor prognosis, with high rates of disease progression and mortality. After realizing the implication of HER2, it first became investigated as a target for treatment in breast cancer, and later expanded to areas of research in other cancer types. To this day, the most therapeutic advancements of anti-HER2 therapy have been in breast cancer; however, there have been strong advancements made in the incorporation of anti-HER2 therapy in other cancer types as well. This comprehensive review dissects HER2 to its core, incorporating the most up to date information. The topics touched upon are discussed in detail and up to 200 published sources from the most highly recognized journals have been integrated. The importance of knowing about HER2 is exemplified by the groundbreaking advancements that have been made, and the change in treatment plans it has brought to the oncological world in the last twenty years. Since its groundbreaking discovery there have been significant breakthroughs in knowledge regarding the actual receptor, the receptors biology, its mechanism of action, and advancements in tests to detect HER2 and significant strides on how to best incorporate targeted treatment. Due to the success of this field thus far, the review concludes by discussing the future of novel anti-HER2 therapy currently in development that everyone should be aware of.

## 1. Introduction

Human epidermal growth factor receptor 2 (HER2) belongs to the ErbB family, a group of four transmembrane glycoproteins with tyrosine kinase activity, all structurally related to the epidermal growth factor receptor (EGFR), its first discovered member. The ErbB family includes EGFR (also termed HER1 or ErbB1), HER2 (ErbB2), HER3 (ErbB3), and HER4 (ErbB4). These tyrosine kinases are involved in the transmission of cellular signals controlling normal cell growth and differentiation. If this transmission goes awry, it can lead to dysregulated growth of the cell. HER2 specifically can be implicated in the pathogenesis of at least eight malignancies with varying prevalences, including breast, gastroesophageal, ovarian, endometrial, bladder, lung, colon and head and neck cancers. HER2 is implicated in 15–20% of breast cancers [1], 15–20% of gastroesophageal adenocarcinomas [2], 8–66% of ovarian cancers [3], 17–80% of endometrial cancers [3], 6–17% of bladder cancers [4], 2–4% of lung cancers [5], 3–5% of colon cancer [6], and 50% of head and neck cancers [7]. The HER2 receptor is encoded by the HER2 gene, a proto-oncogene on chromosome 17q21. It has been found that in some malignancies (breast, gastric, and ovarian), HER2 is overexpressed due to HER2 gene amplification. However, the HER2 gene is not amplified in a few tumor types (lung, bladder and esophageal) and in these cases overexpression may result from transcriptional or post-transcriptional dysregulation [8].

HER2 positivity quickly became a well-characterized indicator of aggressiveness and poor prognosis, with high rates of disease progression and mortality [9,10].

After realizing the implication of HER2, it first became investigated as a target for treatment in breast cancer, and later expanded to areas of research in other cancers types. To this day, the most therapeutic advancements of anti-HER2 therapy have been in breast cancer. Dr. Dennis Slamon led the first landmark clinical trial with the anti-HER2 monoclonal antibody trastuzumab, which showed significant improvement in the rates of disease progression and survival when compared to chemotherapy alone in patients with HER2-positive metastatic breast cancer. This led to trastuzumab’s groundbreaking approval by the Food and Drug Administration (FDA) in 1998 [11]. 

HER2 specifically can be implicated in the pathogenesis of at least eight malignancies. After realizing the implication of HER2, it first became investigated as a target for treatment in breast cancer, and later expanded to areas of research in other cancers types. To this day, the most therapeutic advancements of anti-HER2 therapy have been in breast cancer; however, there have been strong advancements made in the incorporation of anti-HER2 therapy in other cancer types as well. This comprehensive review dissects HER2 to its core, incorporating the most up-to-date information. The importance of HER2 can also be measured by the amount of continued research being conducted, which will continue to implicate cancer treatment plans moving forward. For these many reasons, it is important to have a strong baseline understanding of this very immense topic, which this review will provide to readers. 

## 2. HER2 Discovery 

EGFR (ErbB1/HER1) was the first tyrosine kinase discovered by Carpenter et al. at Vanderbilt University in 1978. The first evidence of ErbB2/HER2 was inferred from the connection to its rat homologous gene: neu. The HER2 gene (neu in mice) was officially discovered between 1982 and 1984 by the Robert A. Weinberg group at the Massachusetts Institute of Technology, Rockefeller University and Harvard University. In 1987, Dr. Dennis Slamon found that HER2 gene amplification was linked to breast cancer in about 20% of cases [12]. This monumental discovery led to further research and subsequent clinical trials in the 1990’s, and eventually to the development of the groundbreaking drug: trastuzumab. This targeted drug was the first of its kind to become the standard treatment for HER2-overexpressed breast cancers. After this discovery, it opened the vast possibility of HER2-directed therapy in various forms. There was hope that targeted therapy could benefit any solid tumor with overexpression of HER2, and it fueled the extensive effort and research that was put towards this topic in the years to follow, up to the present time. 

## 3. HER2 Biology

The normal biology of HER2 signaling is necessary to understand the concept of HER2 as an oncogene and how current targeted treatments work. A range of growth factors serve as ligands, but none are specific for only the HER2 receptor. Different HER receptors exist as both monomers and dimers, either homo- or heterodimers. A ligand binding to HER1, HER3 or HER4 induces rapid dimerization with a preference for HER2 as a dimer partner [8]. HER2-containing heterodimers generate intracellular signals that are significantly stronger than signals from other HER combinations. In normal cells, few HER2 molecules exist at the cell surface, so few heterodimers are formed, and growth signals are controllable. When the HER2 receptor is overexpressed at the surface, this leads to the formation of multiple HER2 heterodimers, resulting in stronger cell signaling and enhanced responsiveness to growth factors and malignant growth. Once malignant growth ensues, HER2 transforms from a proto-oncogene to an oncogene [8]. 

## 4. HER2 as an Oncogene and Overexpression in Cancers

All four HER receptors (HER1/EGFR, HER2, HER3, HER4) are composed of a cysteine-rich extracellular ligand binding site, a transmembrane lipophilic segment, and an intracellular domain with tyrosine kinase catalytic activity [13]. The extracellular domain of HER proteins can exist in a closed inhibited or in an open active conformation. The HER receptors exist as monomers on the cell surface, and ligand binding induces a conformational change in their extracellular domain that induces the active conformation and promotes their dimerization and consequent transphosphorylation [14]. Figure 1 demonstrates the HER2 signaling pathway. 

Unlike the other members of the family, the extracellular domain of HER2 does not pivot between active and inactive conformations and constitutively exists in an activated conformation [15]. Consistent with its constitutively active conformation, HER2 lacks ligand binding activity, and its signaling function is engaged by its ligand-bound heterodimeric partners, such as HER1 and HER3 [16]. The HER2 receptor homo- or heterodimerization results in the autophosphorylation of tyrosine residues within the cytoplasmic domain of the receptors and engages a variety of signaling pathways, principally the mitogen-activated protein kinase (MAPK), phosphatidylinositol-4,5-bisphosphate 3-kinase (PI3K), and protein kinase C (PKC), resulting in cell proliferation, survival, differentiation, angiogenesis, and invasion [17]. Moreover, HER2 dimerization promotes the rapid degradation of cell-cycle inhibitor p27^Kip1^ protein, leading to cell-cycle progression [18]. 

HER2 has the strongest catalytic kinase activity and HER2-containing heterodimers have the strongest signaling functions [19]. The HER2-HER3 heterodimer is the most potent stimulator of downstream pathways, particularly the PI3K/AKT/mTOR pathway, a master regulator of cell growth and survival.

## 5. Testing for HER2

HER2 testing is a standard procedure for all new breast cancer diagnoses, as well as in cases of metastatic tumor progression. It is also common practice to perform HER2 testing in patients with advanced gastroesophageal cancer, colorectal cancer, lung cancer, ovarian cancer, endometrial cancer, bladder cancer, and head and neck cancer. First, we will focus on HER2 testing in breast cancer.

This analysis relies on a combination of immunohistochemistry (IHC) and in situ hybridization (ISH). In particular, IHC detects the expression and intensity of HER2 protein on the cell membrane by a three-tier scoring system (from score 0 to score 3+), while ISH detects the presence of gene amplification using HER2 and CEP17 probes in chromosome 17 [20]. Accurate determination of HER2 status is critical for optimizing therapy and outcomes.

### 5.1. Immunohistochemistry

IHC utilizes labeled antibodies that bind specifically to their target antigens in situ, making it possible to visualize and document the high-resolution distribution and localization of specific cellular components within cells.

The American Society of Clinical Oncology (ASCO) and College of American Pathologists (CAP) have collaboratively laid down guidelines and algorithms for the evaluation of HER2 protein expression by IHC assay of the invasive component of breast cancer specimens (Figure 2). 

### 5.2. In Situ Hybridization

In situ hybridization (ISH) is a technique that allows for the precise localization of a specific segment of nucleic acid within a histologic section. The underlying basis of ISH is that nucleic acids can be detected through the application of a complementary strand of nucleic acid to which a reporter molecule is attached. Detection of the probe can be achieved by chromogenic or fluorescent techniques referred to as chromogenic in situ hybridization or fluorescence in situ hybridization (FISH), respectively.

If IHC results are 2+ or equivocal, it is recommended to perform ISH. ISH can be performed with a single HER2 probe (Figure 3) or dual HER2 and CEP17 probes (Figure 4). CEP17 stands for chromosome enumeration probe-targeting centromere 17, and it serves as a control probe within chromosome 17 for correction of aneuploidy. An expert panel recommended the use of dual-probe instead of single-probe ISH assays. The number of HER2 signals, the number of CEP17 signals, and their ratio are the three parameters for HER2 FISH interpretation by dual-probe ISH assay. An average HER copy number greater than 6.0 signals or a HER/CEP17 ratio of >2.0 is automatically considered positive for HER2 gene amplification [21]. 

### 5.3. Current Challenges with IHC Testing and “HER2-Low” in Breast Cancer

Currently, breast cancer patients with HER2 IHC 3+ and 2+/ISH amplified are considered HER2-positive and are eligible for several therapies that disrupt the HER2 signaling pathway. HER2 with IHC 0, 1+ and IHC 2+/ISH not amplified are considered HER2-negative for protein expression/gene amplification. The degree of accuracy of HER2 testing has become even more pivotal in recent times after the DESTINY-Breast 04 trial, which led to the 2022 FDA approval of fam-trastuzumab deruxtecan (T-DXd) for metastatic breast cancer patients with “HER2-low” expression, that is, IHC expression (1+ or 2+), despite negative ISH amplification [22]. Even though only around 15% of tumors meet the guideline criteria, around 30–60% of the tumors traditionally defined as “HER2-negative” show low levels of HER2 expression, in the absence of gene amplification [23,24]. 

HER2 IHC testing was initially designed to distinguish high levels of HER2 expression (i.e., almost 2 million molecules/cell which corresponds to HER2 IHC 3+) from lower levels of HER2 expression (i.e., 20,000 to 500,000 molecules/cell for HER2 IHC 0 to 2+). Therefore, this method has not been developed for detecting the dynamic range of HER2-low tumors [25]. HER2 expression is dynamic and can change as the disease progresses, with up to 40% discordance between primary and metastatic tumors [26,27], which supports the guideline recommendation to retest HER2 status after progression into metastatic disease. The inherent subjectivity of HER2 IHC assessment and the frequency of intratumoral heterogeneity are other barriers to HER2-low assessment [28,29]. Intratumoral HER2 IHC heterogeneity (i.e., uneven distribution of HER2 expression or different intensities of HER2 staining in tumor cells) is more frequent in HER2-low (2 + or 1 + IHC score) samples [30].

These factors can all lead to varying interpretations of HER2 IHC status in HER2-low samples by different pathologists. In a study by Fernandez et al., the team aimed to assess the concordance of 18 pathologists reading 170 breast cancer biopsies, noting only a 26% concordance rate in IHC values between 0 and 1+ compared with 58% concordance between 2+ and 3+ [31]. Prat et al. showed that there was 77% agreement between historical and centrally rescored HER2-low status [32]. In a global retrospective study conducted by Viale et al., out of 529 historical samples that were rescored and noted to be HER2-low, the rate of concordance for HER2-low status was 81.2%, and more than 30% of historical IHC 0 cases were rescored as HER2-low [33]. 

The challenges in HER2 testing suggests that the current assays for HER2 may need revision for optimal patient care. Moutafi et al. designed an assay to increase the resolution and sensitivity of HER2-low expression in unamplified cases by using quantitative immunofluorescence to test a range of antibody concentrations. The amount of HER2 protein was measured in units of attomols/mm^2^ by mass spectrometry. By calculating the limits of detection, quantification, and linearity of this assay, Moutafi et al. determined an optimal dynamic range of low HER2 expression to be between 2 and 20 attomol/mm^2^, which can potentially serve as a more accurate assessment for HER2-low status and thus candidacy for treatment with ADCs like TDxd [34]. 

More research is warranted in this area to further standardize HER2 testing. 

### 5.4. HER2 Testing in Other Malignancies

HER2 testing in metastatic gastroesophageal adenocarcinoma follows same principles of IHC and ISH testing as breast cancer. As per guidelines from the College of American Pathology and, the American Society of Clinical Oncology, it is recommended to assess for HER2 status in all patients with advanced gastroesophageal cancers. As in breast cancer, the pathologist should perform IHC testing first, followed by ISH when IHC is 2+ (equivocal). Positive (3+) or negative (0 or 1+) HER2 IHC results do not require further ISH testing [35]. 

Diagnostic criteria for HER2 positivity in metastatic colorectal cancer were initially not standardized and were derived from those used in gastric cancer. Valorta et al. proposed more stringent criteria to assess HER2 positivity in metastatic colorectal cancer [36]. Based on these new criteria, HER2 is positive with 3+ IHC overexpression, with intense complete circumferential or lateral staining in more than 50% of tumor cells (versus 10% in breast and gastric). Equivocal cases (IHC incomplete staining with moderate intensity in >50% or complete intense staining in 10–50% of tumor cells) must be analyzed further by ISH. ISH is then considered positive if the HER2-to-CEP 17 ratio is >2.0 in more than 50% of the tumor cells. Patients can only be eligible for HER2-directed therapy if they are HER2-positive and negative for mutations in RAS and BRAF. 

Molecular techniques such as next-generation sequencing and comprehensive genomic sequencing are alternative methods to identify HER2 alterations in metastatic colorectal cancer, which can also make patients candidates for HER2-directed therapy. Next-generation sequencing has the advantage of detecting a wider range of molecular alterations and quantifying the gene copy number [6]. Schrock et al. and Takegawa et al. demonstrated that HER2 mutations and alterations are also detectable in the circulating tumor DNA (ctDNA) of patients with metastatic colorectal cancer and there was molecular concordance between plasmatic ctDNA and tissue samples [37,38].

For patients with metastatic non-small-cell lung cancer, HER2-directed therapy is currently recommended for patients with activating mutations, independent of protein expression, which is not routinely tested. The current recommendation is to test for genetic alterations in ERBB2, the gene that encodes for the HER2 receptor. Mutations in ERBB2 are most commonly insertion/duplication events in exon 20. While some mutations can be activating, including mutations in the extracellular domain and exon 20 insertion/duplication mutations, not all mutations in ERBB2 are activating. Next-generation sequencing is the most commonly used modality to detect ERBB2 mutations. While ISH and IHC testing can detect HER2 amplification and expression, respectively, they are not routinely recommended outside of the context of clinical trials for non-small-cell lung cancer [39]. 

### 5.5. The Future: De-Escalating Therapy with the HER2DX Genomic Tool

HER2DX is a recently developed prognostic and predictive 27-gene genomic assay developed to guide the use of de-escalated HER2-directed therapy in patients with early-stage HER2-positive tumors. While larger trials are needed before the assay and its scores are fully validated for clinical use, current data are promising and discussed here. 

The assay is based on clinical features and the expression of four gene signatures, including ERBB2 mRNA levels [40]. 

The four gene signatures:Immunoglobulin (IGG) signature (14 genes);Tumor cell proliferation signature (4 genes);Luminal differentiation signature (5 genes);HER2 amplicon signature (4 genes).

The HER2DX assay integrates clinical information (i.e., tumor size and nodal status) with biological information from the above gene signatures to provide three independent scores to predict both the long-term prognosis and likelihood of pCR in HER2-positive early breast cancer [41].

The calculated HER2DX scores are

HER2Dx risk score—based on the IGG, the luminal and the proliferation signatures;HER2DX pCR likelihood score—based on HER2, IGG, luminal and proliferation signatures;HER2DX ERBB2 score—based on the ERBB2 mRNA levels.

In the PerELISA trial, Guarneri et al. evaluated the ability of HER2DX to predict the efficacy of a de-escalated, chemotherapy-free neoadjuvant regimen in postmenopausal women with stage II and IIIA HER2-positive/HR-positive breast cancer that is highly estrogen-sensitive. The high degree of estrogen sensitivity was determined by the subjects receiving 2 weeks of neoadjuvant letrozole followed by a >20% drop in Ki67, based on repeat biopsy. A total of 40 out of 55 patients were identified to be highly estrogen-sensitive, and they continued neoadjuvant chemotherapy-free therapy with letrozole and five cycles of trastuzumab and pertuzumab. The primary endpoint was the ability of the three HER2DX scores to predict pathological complete response (pCR). The study suggested that high HER2DX pCR and ERBB2 mRNA scores were both significantly associated with treatment response. Patients with low, medium, and high HER2DX ERBB2 mRNA scores achieved pCR rates of 0%, 7.7% and 53%, respectively. There was no association of the HER2DX prognostic risk score with treatment response.

The DAPHNe phase II trial also tested the validity of the HER2DX assay in predicting response with de-escalated neoadjuvant therapy. In this single-arm trial, 80 patients with stage II to III HER2-positive breast cancer were treated with 12 weeks of paclitaxel, trastuzumab, and pertuzumab. The primary endpoint was the ability of the HER2DX pCR score to predict pCR. There was again a high association, with high, medium, and low scores achieving pCR rates of 92.6%, 63.6% and 29.0%, respectively [42]. 

HER2DX pCR scores could predict pCR rates following neoadjuvant therapy and might guide the selection of patients for a de-escalated neoadjuvant treatment approach. Further research is warranted to fully implement this into clinical practice. 

## 6. HER2 as a Prognostic and Predictive Biomarker

HER2 has both prognostic and predictive implications for invasive breast cancers [43]. Before the era of HER2-directed therapy, HER2 gene amplification was known to be otherwise associated with shorter disease-free and overall survival in breast cancers, as per a 1987 study conducted by Slamon et al. [44]. In another 1993 study by Press et al., the expression of HER2 was evaluated in 704 node-negative breast cancer patients, and it was noted that women with breast cancer and high HER2 overexpression had a risk of recurrence 9.5 times greater than those with normal HER2 expression [45]. HER2-amplified breast cancers also have increased propensity to metastasize to the brain [46]. In another study by Seshadri et al. in 1993, 1056 patients with stage I–III breast cancer were studied and HER2 amplification was again noted to be associated with significantly shorter disease-free survival [47]. 

In gastric cancer, HER2 overexpression is also directly correlated with poorer outcomes. Multiple other studies also note similar findings of HER2 overexpression being a negative prognostic factor, with higher rates of tumor progression and lymph node metastasis [48,49,50,51]. In colorectal cancer, HER2 amplification or overexpression was associated with a lack of response to anti-EGFR therapy. Yonesaka et al. aimed to identify the mechanisms of de novo and acquired cetuximab resistance beyond KRAS in preclinical models. Aberrant activation of HER2 signaling either through HER2 gene amplification or through the overexpression of the HER2-activating ligand led to the persistent activation of the ERK1/2 pathway (extracellular signal-regulated kinase), preventing the cetuximab-mediated growth inhibition that is normally caused by the downregulation of ERK1/2 signaling [52]. HER2 is a well-established negative predictive biomarker in metastatic colorectal cancer, as it hampers the efficacy of anti-EGFR therapy.

In advanced epithelial ovarian cancer, the association of HER2 overexpression with poor survival was first established in a study by Berchuck et al., where they evaluated 73 patients with HER2 overexpression who were found to have significantly worse survival as compared to patients with normal expression. Additionally, patients with high HER2 expression were significantly less likely to have a complete response to primary therapy. 

In endometrial carcinoma, HER2 overexpression and amplification has been linked to poor prognosis. In a study by Santin et al., 30 samples from patients with uterine serous papillary endometrial carcinoma were evaluated for HER2 gene amplification using FISH. The patients with HER2 amplification were found to have a significantly shorter survival time from diagnosis to disease-related death when compared to FISH-negative patients [53].

Since trastuzumab was first discovered in the late 1990s, and with all of the anti-HER2 therapy that has been discovered thereafter, it has further added to the predictive value of HER2, as these therapies have been shown to achieve oftentimes profound responses when there is HER2 positivity. As a result, these therapies have positively and markedly impacted the prognostic landscape of HER2-positive cancers. Today, for instance, with dual HER2-directed therapy in early-stage HER2-positive breast cancers, the 8 year median OS is nearly 93% [54]. 

## 7. Anti-HER2 Therapies

There are currently three different types of anti-HER2-targeted therapy: monoclonal antibodies, tyrosine kinase inhibitors and antibody–drug conjugates. 

### 7.1. Monoclonal Antibodies

Monoclonal antibodies (mAbs) that target HER2 were the first groundbreaking anti-HER2 targeted therapy, discovered in the 1990s. They have a complex mechanism of action and their effects are exerted in multiple ways, including HER2 protein downregulation, the prevention of HER2-containing heterodimer formation, the initiation of G1 cell cycle arrest by induction of the p27 tumor suppressor, the prevention of HER2 cleavage, the inhibition of angiogenesis, and the induction of immune mechanisms [55]. Each one of these mechanisms will be briefly discussed.

HER2 downregulation.Anti-HER2 mAbs reduce the amount of HER2 protein expressed on the cell surface; this is due to the accelerated endocytic degradation of the overexpressed HER2 homo/heterodimer. When the mAb binds to HER2, it has also been shown to inhibit tyrosine auto-phosphorylation of the receptor. The presence of decreased amounts of HER2 on the surface of the cell reduces HER2 homodimerization and stimulatory activity, reversing the transformed phenotype of HER2-overexpressing cells [55].Prevention of heterodimer formation.The formation of heterodimers between HER2 and other members of the HER family is important for the complex control of intracellular signaling and cell growth. HER3 and HER4 preferentially heterodimerize with HER2, and their activity is impaired when HER2 is not present in a cell. Anti-HER2 mAbs interfere with the stability of HER2-HER3 and HER2-HER4 heterodimers, which leads to accelerated ligand dissociation. This leads to decreased cell growth signaling [55].Initiation of G1 cell cycle arrest and induction of p27 tumor suppressor.Anti-HER2 mAbs have anti-proliferative activity, and this effect is cytostatic rather than cytotoxic. This results in an increase in the percentage of cells in the G0/G1 phase, accompanied by a decrease in the percentage of cells in the S phase. Known inhibitors of the cell cycle, such as p27 and p130, are induced when HER2-overexpressing cells are exposed to mAbs. Furthermore, mAbs have cytotoxic effects by sensitizing HER2-overexpressing cells to the tumor necrosis factor-alpha (TNF-alpha), part of the host defense mechanism against tumors [55].Prevention of HER2 cleavage.It has been demonstrated that HER2 extracellular domain levels (ECDL) correlate with a poor prognosis and decreased responsiveness to hormone therapy and chemotherapy [55]. MAbs appear to inhibit HER2 cleavage from HER2-overexpressing cells, which leads to decreased HER2 ECDL. Maintenance of the intact form of HER2 on the cell surface decreases constitutive receptor activation and signal transduction, thereby inhibiting cell growth [55].Inhibition of angiogenesis.Studies have demonstrated that treating HER2-overexpressing breast cancer cells with mAbs can inhibit vascular endothelial growth factor (VEGF) production [55]. VEGF stimulates angiogenesis through intracellular signaling after binding to endothelial cells. This process is controlled by the balance of other signals from angiogenesis inhibitors. In normal homeostasis, there is a balance of both of these signals so that new vessels are created only when they are in need, such as during recovery or growth. Cancer cells can produce pro-angiogenic signals due to an increased demand in blood supply and nutrients. This further allows the cancer cells to grow and invade surrounding tissue, potentially leading to distant metastases.Induction of host immune response.Evidence indicates that mAbs efficiently induce antibody-dependent cellular cytotoxicity against HER2-positive cancer cells but not against cells that do not overexpress HER2 [55]. The fragment C (Fc) portion of mAbs such as trastuzumab can bind to polymorphic receptors on immune cells (natural-killer cells, lymphocytes, macrophages and neutrophils), activating them and enhancing their cytotoxic antitumor activity [56].Anti-Her2 MAbs and chemotherapeutic agents.Various anti-HER2 mAbs have been shown to have synergistic anti-tumor effects when used in combination with chemotherapy. These effects are specific to HER2-overexpressing cells [55]. The higher response rates of trastuzumab when used in combination with chemotherapy (60%) vs. monotherapy (11%) show the importance of the two treatments synergistically working together [57]. This synergistic approach occurs due to DNA damage caused by chemotherapy, while mAbs help to block DNA repair in HER2+-overexpressing cells. It has also been shown that when paclitaxel is given prior to trastuzumab, antibody-dependent cellular toxicity is significantly enhanced, with the rapid recruitment of natural killer cells [58].

Trastuzumab and pertuzumab are two specific groundbreaking monoclonal antibodies that have significantly impacted the treatment landscape of HER2-positive cancers. Trastuzumab, first approved by the FDA in 1998, works by binding to the extracellular domain (ECD IV) of the HER2 receptor, inhibiting downstream signaling pathways and inducing antibody-dependent cell-mediated cytotoxicity (ADCC) [59,60]. Based on preclinical data demonstrating synergy between cytotoxic agents and trastuzumab, clinical trial designs with chemotherapy combinations demonstrating improved survival helped establish trastuzumab as the standard-of-care treatment for both metastatic and early-stage HER2-positive breast cancer. Trastuzumab has since gained indications for the treatment of patients with metastatic, gastric, and colorectal cancers (see Table 1). Pertuzumab, first FDA-approved in 2012, targets ECD II on the HER2 receptor, and can act synergistically with trastuzumab by preventing the heterodimerization of HER2 with other HER receptors such as HER3, resulting in further inhibition of downstream tumor signaling [61]. The addition of pertuzumab to trastuzumab has been shown to augment therapeutic benefit by blocking HER2/HER3 signaling [62]. Given their complementary mechanisms of action at the HER-receptor level, their effect on immune-mediated anti-tumor activity, and their complement-mediated cytotoxicity, the combination of these two agents is thought to be synergistic [63,64].

Margetuximab is the newest moncoloncal antibody targeting HER2, which began to be used in 2020. Margetuximab binds the identical epitope of HER2 receptor; however, it has a much stronger affinity. This difference is due to a replacement of five amino acids in the IgG1 Fc domain, which leads to the improved ADCC of margetuximab. Maregetuximab is able to maintain trastuzumab’s antiproliferative effects, while also enhancing the activation of innate and adaptive immune responses [85].

### 7.2. Tyrosine Kinases

HER2 is a transmembrane glycoprotein with tyrosine kinase activity, in which the phosphorylation of tyrosine residues in the cytoplasmic domain of the receptor molecule lead to downstream cellular growth-promoting pathways. Tyrosine kinase inhibitors (TKIs) specific to HER2 work by competing with ATP for binding at the HER2 catalytic kinase domain, thereby blocking HER2 signaling. Most of these compounds target more than one HER receptor, which has the potential advantage of simultaneously blocking two or more heterodimer components [86]. This leads to an accumulation of inactive receptors at the cell surface. This accumulation at the surface also enhances immune mediated mAb-dependent cytotoxicity [87]. Some advantages of TKIs are their small molecular size, their oral bioavailability, and their ability to cross the blood–brain barrier, which could be important in cases of metastatic disease to the brain. Of note, TKIs have the ability to inhibit multiple kinases, which can be clinically useful with the simultaneous blockage of multiple growth-promoting pathways. On the other hand, there is also a risk that inhibition of multiple pathways could lead to greater toxicity. This highlights the importance of TKIs with higher specificity towards HER2 [88]. 

The first anti-HER2 TKI was lapatinib, a reversible TKI first FDA-approved in 2007 in conjunction with capecitabine for patients who progressed on mAb-based therapy. Three years later, lapatinib was approved with the use of letrozole as a first-line therapeutic option for triple-positive breast cancer [89]. In contrast, neratinib is a second-generation, irreversible pan-HER TKI that targets EGFR, HER2 and HER4, which leads to a greater effect than lapatinib but also higher toxicity, with the most common dose-limiting adverse event being diarrhea. Neratinib was first FDA-approved in 2017 for extended adjuvant treatment in early-stage HER2-positive breast cancer [90]. 

Tucatinib is a third-generation reversible highly selective anti-HER2 TKI that has >1000-fold greater potency for HER2 than EGFR, with diarrhea, nausea, hand–foot syndrome, and fatigue as the most common adverse effects. Tucatinib has also demonstrated high levels of penetration in the central nervous system (CNS). It was first FDA-approved in 2020 for the treatment of HER2-positive metastatic breast cancer, including in patients with CNS metastases [91].

Pyrotinib is another irreversible pan-HER TKI that targets EGFR, HER2 and HER4. Pyrotinib in combination with capecitabine demonstrated good tolerability and antitumor activity and was approved in China in 2018 for the treatment of advanced or metastatic breast cancer treated with prior trastuzumab and taxane [92]. Similar to the other pan-HER2 TKIs, pyrotinib has diarrhea as its most common toxicity. Several trials with pyrotinib are ongoing in breast and other cancers, but it is currently not approved in other countries [91].

### 7.3. Antibody–Drug Conjugates

An antibody–drug conjugate (ADC) is composed of an antibody, a linker and a cytotoxic payload. Humanized or chimeric immunoglobulin G (IgG) is the most commonly used antibody backbone. Payloads are cytotoxic agents that are highly potent and delivered directly into the tumor cells via antibody-mediated endocytosis, thereby achieving more accurate and effective cytotoxicity. The linker binds the payload to the antibody and must be stable in circulation to deliver the payload directly to the tumor cells and avoid its premature release in the bloodstream [93]. Second- and third-generation ADCs have achieved improved linker stability and therefore improved toxicity profiles. 

After binding of the ADC-HER2 complex, it is internalized in an endosome and transported to lysosomes. Once the payload is released, it elicits antitumor activity within targeted cells. Depending on the linker and payload combination, the payload can be released within the extracellular space before or after the ADC internalization, where it can also exert its activity in the neighboring cells, which may or may not express HER2. This wider drug delivery to tumor cells, known as the “bystander effect”, has markedly improved the activity of ADCs in cancer with heterogenous and/or low HER2 expression [93]. 

Ado-trastuzumab-emtansine (TDM-1), a second-generation ADC, showed improvement in overall survival in second- and third-line settings in metastatic HER2-positive breast cancer when compared to lapatinib and capecitabine, following the failure of trastuzumab and taxane [94]. It was first approved by the FDA for HER2-positive metastatic breast cancer in 2013, and subsequently in 2019 it was approved in the adjuvant setting for early-stage HER2-positive breast cancer with residual invasive disease after neoadjuvant chemotherapy. The most common adverse effect with TDM-1 are fatigue, thrombocytopenia, increased aminotransferase levels, and neuropathy. 

Fam-trastuzumab deruxtecan-nxki (T-DXd), a third-generation ADC that exhibits the potent bystander effect, was shown to be significantly superior to TDM-1 in previously treated HER2-positive metastatic breast cancer [95]. T-DXd was first FDA-approved for this indication in 2019, and it has since also been approved for HER2-low metastatic breast cancer, HER2-positive advanced gastric cancer, and HER2-mutant metastatic non-small-cell lung cancer. Some of the most common reported adverse effects with T-DXd include neutropenia, anemia, and nausea. It was also associated with an increased risk of ILD.

The three drug categories discussed above have been extensively studied over the years, with more research on the horizon. Most of the research originated in breast cancers and has since expanded to other solid tumors. It is important to have a basic understanding about several of the trials that have incorporated targeted anti-HER2 therapy in various solid tumors and how they changed the landscape of oncology as we know it. 

The evolution and use of anti-HER2 mAbs, TKIs and ADCs will first be discussed in regard to breast cancer. 

## 8. Anti-HER2 Therapy—Landmark Trials in Breast Cancer

### 8.1. Monoclonal Antibodies

The first groundbreaking trial for anti-HER2 therapy was with trastuzumab and was published in 2001 by Dr. Dennis Slamon, although preliminary trial results had previously led to the FDA approval of this drug in 1998. This study evaluated the efficacy and safety of trastuzumab in patients with HER2-positive metastatic breast cancer, who were randomly assigned to receive chemotherapy (CT) alone vs. CT and trastuzumab. The CT backbone was either doxorubicin and cyclophosphamide, or paclitaxel if the patient had previously received anthracycline in the adjuvant setting. The addition of trastuzumab to CT was associated with a longer progression-free survival (PFS) (median, 7.4 vs. 4.6 months; *p* < 0.001), a higher rate of objective response (50 percent vs. 32 percent, *p* < 0.001), a longer duration of response (median, 9.1 vs. 6.1 months; *p* < 0.001), a lower rate of death at 1 year (22 percent vs. 33 percent, *p* = 0.008), longer overall survival (OS) (median survival, 25.1 vs. 20.3 months; *p* = 0.046), and a 20-percent reduction in the risk of death [11].

Trastuzumab was then FDA approved in 2006 in the adjuvant setting for early-stage HER2-positive breast cancer after a joint analysis of two phase 3 trials comparing adjuvant CT with or without 52 weeks of trastuzumab **(NSABP trial B-31 and NCCTG trial N9831).** The CT backbone was four cycles of doxorubicin and cyclophosphamide followed by paclitaxel for 12 weeks (AC-T). A total of 2043 patients were enrolled in B-31 and 1633 patients were enrolled in N9831. Out of 394 total events (including cancer recurrence, second primary cancer, and death), only 133 events were observed in the trastuzumab-CT group, while 261 events were observed in the CT-only group (HR 0.48, *p* < 0.0001). There was a 33% reduction in the risk of death with the addition of trastuzumab. The risk of cardiac events was 4.1% in the B-31 trial and 2.9% in the N9831 trial [96].

After noting the cardiac toxicity of regimens containing both anthracyclines and trastuzumab, efforts were made to develop non-anthracycline regimens. Based on preclinical synergies between trastuzumab and platinums and docetaxel that were not observed with anthracyclines or paclitaxel, the **BCIRG-006 trial** chose to study the combination of docetaxel, carboplatin, and trastuzumab (TCH). In this study, 3222 patients with early-stage HER2-positive breast cancer were randomized to receive adjuvant CT with doxorubicin and cyclophosphamide followed by docetaxel (AC-T), the same CT regimen plus 52 weeks of trastuzumab (AC-TH), or docetaxel and carboplatin plus 52 weeks of trastuzumab (TCH). The primary endpoint was disease-free survival (DFS) and the secondary endpoints were overall survival (OS) and safety. The 5-year DFS rates were 75% for AC-T, 84% for AC-TH (HR 0.64; *p* < 0.001), and 81% for TCH (HR 0.75; *p* = 0.04). The 5-year rates of overall survival were 87% for AC-T, 92% for AC-TH (HR 0.63; *p* < 0.001), and 91% for TCH (HR 0.77; *p* = 0.04). In contrast, no significant difference in the rate of disease-free or overall survival was seen between the two trastuzumab-containing regimens, but both combinations showed that trastuzumab improved DFS and OS. In terms of safety, the non-anthracycline regimen was preferred due to lower cardiovascular toxicity and a lower risk of leukemia [97].

The **HERA trial** proved that 1 year of adjuvant therapy with trastuzumab was superior to 6 months, and 2 years had no additional benefit compared to 1 year [98]. The **PHARE, HORG, and PERSEPHONE** trials followed to determine if 6 months were non-inferior to one year [98]. While the first two trials failed to prove non-inferiority of 6 months compared to one year of trastuzumab, the **PERSEPHONE** trial showed that 4-year DFS was 89.8% in the 1-year arm vs. 89.4% in the 6-month arm, and OS was 94.8% vs. 93.8%, respectively. However, it was found that the PERSEPHONE trial included a higher-than-average proportion of low-risk node-negative patients (59%) and ER-positive patients (69%), and that the majority of these patients (90%) were treated with a historical anthracycline-containing CT regimen, limiting the applicability of this study in the current non-anthracycline CT practices. While the guideline standard-of-care recommendation remains one year of adjuvant trastuzumab therapy, the results from PERSEPHONE could be considered for patients with limited tolerance to trastuzumab and small (<2 cm), node-negative, ER-positive tumors [99].

Pertuzumab was first FDA-approved for first-line metastatic HER2-positive breast cancer in 2012, shortly after results from the groundbreaking **CLEOPATRA** phase III trial, which compared treatment with docetaxel and trastuzumab with or without pertuzumab. The trial results met the primary endpoint of progression-free survival (PFS). The addition of pertuzumab led to a 6 month improvement in PFS (HR 0.68; 95% CI, 0.58 to 0.80). The updated analysis also showed OS benefit, with the group receiving the pertuzumab combination achieving a median OS of 56.5 months compared with 40.8 months in the group receiving the placebo combination (HR 0.68; 95% CI, 0.56 to 0.84; *p* < 0.001), a difference of 15.7 months [100].

Pertuzumab was then FDA-approved in the neoadjuvant setting in 2013 for early-stage HER2-positive breast cancer (T2 and/or N1), after the **NeoSphere** phase II trial results. This trial compared four groups of neoadjuvant therapy, which was followed by surgery and then further adjuvant anthracycline-based CT with conventional trastuzumab. Neoadjuvant therapy involved four cohorts with varying combinations of docetaxel (T) and trastuzumab (H) with or without pertuzumab (P): TH, THP, HP, or TP. The primary endpoint was pathological complete response (pCR) rates, and secondary endpoints were PFS and DFS. The study showed that the addition of pertuzumab to neoadjuvant CT led to a statistically significant and clinically meaningful 16.8% increase in pCR, with THP achieving pCR in 45.8% of patients, compared to TH’s 29.0% pCR rate. These results also supported the association between pCR and improvements in long-term outcomes. Five-year PFS rates were 81% for TH, 86% for THP, 73% for HP, and 73% for TP (HR 0.69 [95% CI 0.34–1.40] for THP vs. TH). DFS was 81% for TH, 84% for THP, 80% for HP, and 75% for TP. THP vs. TH (HR 0.69 [95% CI 0.34–1.40]). Patients who achieved total pCR (all groups combined) had a longer median PFS of 85%, compared with patients who did not achieve pCR, with a median PFS of 76% (HR 0.54 [95% CI 0.29–1.00]) [100].

The **TRYPHAENA** phase II trial aimed to assess safety tolerability of combined neoadjuvant anti-HER2 therapy with various CT regimens in the treatment of HER2-positive early breast cancer. This trial had a particular focus on cardiac safety, given the known cardiac toxicity of anti-HER2 mAbs and anthracyclines. There were three arms with neoadjuvant CT + HP followed by surgery, then the completion of one year of trastuzumab monotherapy. Two arms were anthracycline-based CT with HP: 5-fluorouracil + epirubicin + cyclophosphamide (FEC) × 3, followed by THP × 3, or concurrent FEC-HP × 3 followed by THP × 3. There was a third non-anthracycline arm with docetaxel + carboplatin + trastuzumab + pertuzumab (TCHP) × six cycles. The trial concluded that the overall combination of CT + HP was safe and tolerable, with low rates of symptomatic left ventricular systolic dysfunction (LVSD) across the study: 5.6% for FEC-HP followed by THP, 4.0% for FEC followed by THP, and 2.6% for TCHP. The pCR rates were also reported, and interestingly the non-anthracycline arm with TCHP had a significant pCR rate of 66.2% (compared to 61.6% and 57.3% for the two anthracycline-based arms), making TCHP one of the clinically preferred neoadjuvant regimens for early-stage HER2-positive breast cancer that is at least 2 cm or lymph node-positive [101].

The **APHINITY** phase III trial studied the addition of pertuzumab to trastuzumab (HP) in the adjuvant setting. Eligible patients included those with early-stage HER2-positive breast cancer, either node-positive or high-risk node-negative (T1c or greater, grade 3, ER-negative, or younger than age 35). After surgery, 4805 patients were randomized to receive standard adjuvant CT with 1 year of trastuzumab, plus or minus pertuzumab. The primary endpoint was invasive disease-free survival (IDFS), and a secondary endpoint was OS. The most updated results were presented in 2022, 8 years after the original trial, which proved that there was a benefit with the addition of pertuzumab. IDFS was 88.4% with pertuzumab vs. 85.8% without pertuzumab, respectively (HR 0.77; 95% CI = 0.66–0.91), amounting to a 2.6% absolute IDFS benefit. While there was a small numerical improvement in OS, it was not statistically significant. The 8-year OS was 92.7% in the pertuzumab group, versus 92.0% in the placebo group, a 0.7% difference (HR 0.83; 95% [CI] = 0.68–1.02, *p* = 0.78). The node-positive cohort clearly derived the most benefit, with an absolute IDFS benefit of 4.9% [86.1% vs. 81.2%] HR 0.72 (95% CI 0.60–0.87), and an absolute benefit of 1.9% for OS. Node-negative patients did not show a benefit, with a IDFS HR of 1.01 and more than 92% of patients being event-free in both arms at 8 years. It was also shown that that both hormone-receptor-negative and hormone-receptor-positive patients benefited from the addition of pertuzumab. Hormone-receptor-negative IDFS had a HR of 0.82 (95% CI 0.64–1.06), and hormone-receptor-positive IDFS had a HR of 0.75 (95% CI 0.61–0.92). Of note, the benefit in hormone-receptor-positive patients was not initially seen in the original trial results, highlighting the importance of long-term follow up to observe a benefit in hormone-receptor-positive patients. In general, long-term survival was excellent for all groups overall, with more than 92% of all patients still alive as of 2022 [54].

The larger trials mentioned thus far mostly focused on Stage II and III HER2-positive breast cancer patients. Due to the paucity of data for Stage I HER2-positive patients, who had been shown to have more than a minimal risk of recurrence and without any standard treatment recommendation at the time, researchers conducted a single-cohort prospective trial for small HER-positive tumors using an abbreviated regimen with paclitaxel and trastuzumab. The **APT trial**, originally published in 2015, focused on the de-escalation of therapy for patients with lower risk of recurrence (3 cm or less, node-negative), with weekly paclitaxel and trastuzumab (TH) for 12 weeks, followed by 9 months of trastuzumab monotherapy. It met its primary endpoint of IDFS. The latest update occurred March 2023, showing that 10-year IDFS was 91.3% (95% CI 88.3–94.4), 10-year recurrence-free interval was 96.3% (95% CI 94.3–98.3), 10-year OS was 94.3% (95% CI 91.8–96.8), and 10-year breast cancer-specific survival was 98.8% (95% CI 97.6–100). This trial also included patients with T1a and T1b tumors (<1 cm), whom had been largely underrepresented in previous trials. It paved the way for treatment of small HER2-positive tumors with a lower toxicity profile, and is currently endorsed by the NCCN guidelines for T1N0 patients, with consideration for tumors as small as 1 mm [102].

In 2020, margetuximab, a new anti-HER2 monoclonal antibody, was FDA-approved after the findings of the **SOPHIA** phase III trial. In this trial, 526 patients with HER2-positive metastatic breast cancer who had previously received at least two lines of anti-HER2 therapy were randomized to receive either margetuximab + CT or trastuzumab + CT. Primary endpoints were PFS and OS. The trial findings showed that the median PFS was 5.7 months with margetuximab + CT and 4.4 months with trastuzumab + CT (HR 0.71; 95% CI, 0.58–0.86; *p* < 0.001). OS in the interim also showed some numerical improvement without reaching statistical significance, with median OS of 21.6 months in the margetuximab group versus 19.8 months in the trastuzumab group (HR, 0.89; 95% CI, 0.69–1.13; *p* = 0.33). Margetuximab also had an acceptable safety profile that was very similar to trastuzumab, with the most common adverse events including fatigue, nausea, diarrhea and neutropenia in both groups with more vomiting in the margetuximab group and more anemia in the trastuzumab group [103].

### 8.2. Antibody–Drug Conjugates

The **EMILIA** phase III trial led to the first FDA approval of trastuzumab emtansine (T-DM1) in breast cancer in 2013. It studied women with HER2-positive metastatic breast cancer previously treated with trastuzumab and a taxane, and randomized them to receive T-DM1 or lapatinib plus capecitabine. The primary endpoints were PFS, OS and safety. T-DM1 drastically improved PFS and OS with less toxicity than lapatinib plus capecitabine. The median PFS was 9.6 months with T-DM1 versus 6.4 months with lapatinib plus capecitabine (HR 0.65; 95% CI 0.55 to 0.77; *p* < 0.001), and median OS at the second interim analysis crossed the stopping boundary for efficacy (30.9 months vs. 25.1 months; HR 0.68; 95% CI 0.55 to 0.85; *p* < 0.001). Rates of adverse events of grade 3 or above were higher with lapatinib plus capecitabine than with T-DM1 (57% vs. 41%). The incidences of thrombocytopenia and increased serum aminotransferase levels were higher with T-DM1, whereas the incidences of diarrhea, nausea, vomiting, and palmar–plantar erythrodysesthesia were higher with lapatinib plus capecitabine [94].

The **KATHERINE** phase III trial was initiated because we know that HER2-positive early breast cancer patients who receive neoadjuvant HER2-based chemotherapy and have residual invasive disease on surgical pathology have a worse prognosis than those who achieve a pathologic complete response [12]. This trial aimed to determine if adjuvant T-DM1 would provide benefit when there is pathological residual disease. HER2-positive early breast cancer patients with residual invasive disease in the breast or axilla at surgery after receiving neoadjuvant therapy containing a taxane (with or without anthracycline) and trastuzumab were randomly assigned to receive adjuvant T-DM1 or trastuzumab for 14 cycles. The primary endpoint was IDFS at 3 years, which was significantly higher in the T-DM1 group, at 88.3%, compared to the trastuzumab group, at 77.0% (HR 0.50; 95% CI 0.39 to 0.64; *p* < 0.001). Adjuvant T-DM1 is now the standard of care for HER2-positive patients with residual disease on pathology [104].

Fam-trastuzumab-deruxtecan-nxki (T-DXd) first received accelerated approval by the FDA in 2019 after results from the **DESTINY-Breast01** phase II trial, which showed T-DXd achieved an overall response rate of 60.9% and a median PFS of 16.4 months in a single cohort of patients with HER2-positive metastatic breast cancer who had previously received at least two anti-HER2 regimens. This PFS exceeded that of any anti-HER2 therapy existing at the time, and paved the way for a head-to-head comparison of T-DXd with T-DM1. In the subsequent **DESTINY-Breast03** phase III trial, a total of 524 patients with HER2-positive unresectable or metastatic breast cancer previously treated with trastuzumab and a taxane were randomly assigned to T-DXd or T-DM1 as second-line therapy or beyond. The primary endpoint was 12-month PFS, which was 75.8% with T-DXd and 34.1% with T-DM1 (HR 0.28; 95% CI, 0.22 to 0.37; *p* < 0.001). A secondary endpoint was 12-month OS, which was 94.1% with T-DXd and 85.9% with T-DM1 (HR 0.55; 95% CI, 0.36 to 0.86; prespecified significance boundary not reached). The incidence of adverse events of any grade was 98.1% with T-DXd and 86.6% with T-DM1. The most common adverse events with T-DXd were bone marrow suppression, transaminitis, nausea, diarrhea, alopecia, and fatigue. An important adverse event of T-DXd was interstitial lung disease or pneumonitis, which occurred in 10.5% of the patients; none of these events were of grade 4 or 5. Overall, a manageable safety profile of T-DXd was confirmed with longer treatment duration [95]. T-DXd is currently the second-line standard-of-care treatment for metastatic HER2-positive breast cancer, after THP.

Due to the potent bystander effect of drugs like T-DXd, which delivers its cytotoxic payload not only intracellularly but also to neighboring cells, efforts were made to explore drug efficacy in breast tumors with lower levels of HER2 expression (“HER2-low”). These cancers would not traditionally be considered to be HER2-positive based on current criteria, that is, they are either hormone-receptor-positive or triple-negative breast cancers. The **DESTINY-Breast04** phase III trial studied patients with HER2-low metastatic breast cancer who had previously received at least one line of chemotherapy in the metastatic setting or who developed disease recurrence within six months of completing adjuvant chemotherapy. A total of 88.7% of all the patients were hormone-receptor-positive (a proportion that is representative of such disease in the HER2-low population), and they must have also received at least one line of endocrine therapy. HER2-low status was defined as a score of 1+ on IHC or as an IHC score of 2+ and negative results on FISH. A total of 557 patients were randomly assigned in a 2:1 ratio to receive T-DXd or chemotherapy (physician’s choice). The primary endpoint was PFS in the hormone-receptor-positive cohort. Secondary endpoints were PFS among all patients and OS in the hormone receptor positive group. There was significantly longer PFS and OS with T-DXd compared to chemotherapy. In the hormone-receptor-positive cohort, the median PFS was 10.1 months in the T-DXd group and 5.4 months in the chemotherapy group (HR 0.51; *p* < 0.001), and OS was 23.9 months and 17.5 months, respectively (HR 0.64; *p* = 0.003). Among all patients, the median PFS was 9.9 months in the T-DXd group and 5.1 months in the chemotherapy group (HR 0.50; *p* < 0.001), and overall survival was 23.4 months and 16.8 months, respectively (HR 0.64; *p* = 0.001) [22,95]. The FDA approved T-DXd in 2022 for patients with HER2-low metastatic breast cancer after having received at least one line of chemotherapy in the metastatic setting or with recurrence within six months of adjuvant chemotherapy.

### 8.3. Tyrosine Kinase Inhibitors

The first anti-HER2 TKI that was discovered was lapatinib (reversible). In 2006, a phase III trial randomly assigned 324 women with HER2-positive metastatic breast cancer previously treated with trastuzumab to treatment with lapatinib-plus-capecitabine versus capecitabine alone. The primary endpoint was median time to progression, which was 8.4 months in the combination-therapy group as compared with 4.4 months in the monotherapy group (HR 0.49 (95% CI 0.34 to 0.71; *p* < 0.001). There were no symptomatic cardiac events and most adverse events were manageable and included diarrhea and hand–foot syndrome [105].

Lapatinib was FDA-approved in 2007 for previously treated metastatic HER2-positive cancer.

Neratinib, a pan-HER2 irreversible TKI, was first studied at a large scale as an extended adjuvant therapy in the **ExteNET** trial. This phase III trial studied neratinib in patients with early-stage HER2-positive breast cancer; while initially all patients with stages 1–3 were eligible, a protocol amendment 7 months into the study restricted eligibility to higher-risk patients with stage 2–3, and patients who completed neoadjuvant chemotherapy were only eligible if there was pathological residual invasive disease at the time of surgery. After completion of neoadjuvant or adjuvant CT with one year of trastuzumab, a total of 2840 patients were randomized to receive neratinib or placebo for one year. The primary endpoint was IDFS, which was indeed significantly improved, particularly in the HER2-positive/hormone receptor-positive (HR+) population. Overall, the IDFS at 5 years was 90.2% in the neratinib group and 87.7% in the placebo group, with 116 versus 163 IDFS events, respectively (HR 0.73, 95% CI 0.57–0.92, *p* = 0.0083). There was lower benefit in T1 (2 cm or less) or node-negative tumors [106]. A consistent finding in the first two interim analyses was that the benefit was more profound in two predefined subgroups: patients who initiated neratinib within one year of trastuzumab completion (versus greater than one year), and patients with HER2-positive/hormone-receptor-positive (HR+) disease (versus hormone-negative). There were 1334 patients who were HER2+/HR+ and initiated neratinib within the one-year window, and for them the absolute IDFS benefit at 5 years was 5.1% (HR 0.58; 95% CI 0.41–0.82). Within this HR+ cohort initiating therapy within one year, an even deeper benefit of was seen in the 295 patients who had residual invasive disease: there was a 7.4% IDFS benefit at 5 years (HR 0.60; 95% CI 0.33–1.07, *p* = 0.086) and a 9.1% absolute OS benefit at 8 years (HR 0.47; CI 0.23–0.92, *p* = 0.03) [107]. The additional benefit in the HR-positive population (most of whom received concomitant endocrine therapy) is thought to be from the inhibition of reciprocal crosstalk between the signaling pathways of the HER2 and estrogen receptors, a known mechanism of resistance for these tumors. A final analysis for OS revealed that in the overall HER2+/HR+ cohort, 8-year overall survival rates were 93.2% in the neratinib group and 90.4% in the placebo group (HR 0.65; 95% CI 0.41–1.03). On the other hand, OS at 8 years was comparable in the intention to treat the population with 90.1% in the neratinib group and 90.2% in the placebo group (HR 0.95; 95% CI 0.75–1.21; *p* = 0.69).

While diarrhea was the most prominent side effect, with grade 3 diarrhea occurring in up to 40% of patients, it was found that either dose escalation or prophylaxis with loperamide/colestipol can significantly improve tolerability [108]. The FDA first approved neratinib in 2017 as an extended adjuvant therapy following the completion of one year of trastuzumab for early-stage HER2-positive breast cancer, based on the trial’s intention to treat the population, whereas the European Medicines Agency approved it only for HER2-positive early-stage breast cancer that is also HR+ and when it is started within 1 year of trastuzumab completion.

The **NALA** phase III trial studied 621 patients with metastatic HER2-positive breast cancer who had received more than two previous HER2-directed therapies, and randomized them to treatment with neratinib plus capecitabine (N + C) versus lapatinib with capecitabine (L + C). The study included patients with stable, asymptomatic CNS disease. The co-primary endpoints were PFS and OS, and the study was considered positive if at least one endpoint was met. Some secondary endpoints were the time to intervention for CNS disease, the overall response rate (ORR), and the duration of response. The results showed that N + C significantly improved PFS and reduced the number of interventions needed for CNS disease compared to L + C. The median PFS was 8.8 months with N + C and 6.6 months with L + C (HR 0.76; 95% CI 0.63 to 0.93; *p* = 0.0059). The median OS was 24.0 months with N + C and 22.2 months with L + C (HR 0.88 (95% CI, 0.72 to 1.07; *p* = 0.2098). The cumulative number of interventions for CNS disease was 22.8% with N + C versus 29.2% with L + C (*p* = 0.043). The ORR was 32.8% with N + C and 26.7% with L + C (*p* = 0.1201), with a median duration of response 8.5 versus 5.6 months, respectively (HR 0.50; 95% CI 0.33 to 0.74; *p* = 0.0004) [90]. The FDA approved neratinib in combination with capecitabine in 2020 for HER2-positive metastatic breast cancer patients previously treated with at least two anti-HER2-based regimens in the metastatic setting.

The **HER2CLIMB** trial investigated tucatinib, a highly selective anti-HER2 TKI. A total of 612 patients who were previously treated with trastuzumab, pertuzumab and T-DM1 were randomly assigned to receive either tucatinib or a placebo in combination with trastuzumab and capecitabine. Notably, patients with brain metastases were also included. The primary endpoint was PFS, and secondary endpoints included OS, PFS among patients with brain metastases, and safety. The study met its primary and secondary endpoints. It was especially found to be helpful in the setting of patients with brain metastatic disease. PFS at 1 year was 33.1% in the tucatinib–combination group and 12.3% in the placebo–combination group (HR 0.54; 95% CI, 0.42 to 0.71; *p* < 0.001), and the median PFS was 7.8 months and 5.6 months, respectively. OS at 2 years was 44.9% in the tucatinib–combination group and 26.6% in the placebo–combination group (HR 0.66; 95% CI, 0.50 to 0.88; *p* = 0.005), and the median OS was 21.9 months and 17.4 months, respectively. Among the patients with brain metastases, PFS at 1 year was 24.9% in the tucatinib–combination group and 0% in the placebo–combination group (HR 0.48; 95% CI, 0.34 to 0.69; *p* < 0.001), and the median CNS PFS was 7.6 months and 5.4 months, respectively. In terms of adverse effects, there was an increased risk of diarrhea and transaminitis in the tucatinib–combination when compared to the placebo–combination group [109].

The **COMPASS-HER2** trial (NCT04266249) is currently studying adjuvant T-DM1 in combination with tucatinib versus placebo for early-stage, high-risk HER2-positive breast cancer patients with residual disease after neoadjuvant HER2-directed therapy. The primary objective of this ongoing phase III trial will be IDFS, and secondary objectives will be breast cancer-free survival, distant recurrence-free survival, brain metastasis-free survival, and overall survival.

## 9. Targeting HER2 across Cancers Other Than Breast

Alterations in HER2 family members play an important role in the development and progression of several human cancers. Breast, gastric and urothelial cancers have been shown to have the highest rates of HER2 overexpression [110]. Here, we will discuss selected important trials of anti-HER2 therapy in cancers other than breast and refer to Table 1 for summary. Table 2 summarizes HER2 alterations which have been evaluated for targetability across various cancer types. 

### 9.1. Gastric and Gastroesophageal Junction Cancers

HER2 amplification or overexpression is the driving force in the development of cancers in approximately 15–20% of gastric or gastroesophageal junction (GEJ) adenocarcinomas [2]. The specific prevalence depends on tumor location and histologic subtype. In the Trastuzumab for Gastric Cancer (ToGA) trial, the overall HER2 amplification or overexpression was 22.1%, and otherwise highest in patients with intestinal histology (31.8%) and the lowest in those with diffuse histology (6.1%), while it was higher in adenocarcinomas located in the GEJ compared to those from the gastric region (32.2% vs. 21.4%, respectively) [111]. 

The phase III randomized **ToGA trial** investigated trastuzumab as a first-line treatment in combination with chemotherapy (fluoropyrimidine and cisplatin) compared to chemotherapy alone in patients with locally advanced or metastatic HER2-positive gastric or GE junction tumors [65]. A total of 584 patients were randomized to both arms. Patients were eligible if they had IHC 3+ or FISH-positive (HER2:CEP17 ratio ≥ 2) tumors [65]. Median overall survival (OS) was 13.8 months in the combination arm vs. 11.1 months in the chemotherapy-alone arm (HR: 0.74, 95% CI 0.60–0.91; *p* = 0.0046) [65]. The greatest effect was observed in patients with high HER2 expression (IHC 2+ and FISH-positive or IHC 3+), with an OS of 16.0 months in the combination arm vs. 11.8 months in chemotherapy-alone arm (HR: 0.65, 95% CI 0.51–0.83; *p* = 0.036) [65]. Less effect was observed in the low-HER2-expression group (IHC 0 and FISH-positive or IHC 1+ and FISH-positive), with 10.6 months in the combination arm vs. 7.2 months in the chemotherapy-alone arm (HR: 0.92, 95% CI 0.48–1.76) in the IHC 0 and FISH-amplified tumors compared to 8.7 months vs. 10.2 months (HR: 1.24, 95% CI 0.70–2.20) in the IHC 1+ and FISH-amplified tumors [65]. Based on these data, trastuzumab plus chemotherapy was FDA-approved for patients with previously untreated, HER2-positive, metastatic gastric or GEJ adenocarcinoma on 20 October 2010 [112]. 

However, the addition of pertuzumab to trastuzumab and chemotherapy has not been shown to add a statistically significant benefit. In the phase III randomized **JACOB trial**, 780 patients with HER2-positive metastatic gastric or GEJ cancers were randomized to receive first-line trastuzumab and chemotherapy with either pertuzumab or a placebo. The primary endpoint was OS. The median OS was 17.5 months in the pertuzumab-containing group versus 14.2 months in the placebo-containing group (HR: 0.84, 95% CI 0.71–1.00; *p* = 0.057) [69].

Another phase IIIb randomized **HELOISE trial** confirmed that loading-dose trastuzumab (8 mg/kg) followed by high-dose maintenance (10 mg/kg every 3 weeks) did not add additional efficacy to standard-dose trastuzumab (loading dose of 8 mg/kg followed by 6 mg/kg maintenance dose every 3 weeks) with chemotherapy [113]. 

Trastuzumab was also FDA-approved in combination with pembrolizumab and chemotherapy for metastatic HER2-positive gastric or GEJ cancers after results from the phase III **KEYNOTE-811 trial**, in which a total of 434 patients with previously untreated metastatic HER2-positive gastric or GEJ adenocarcinoma were randomized to receive pembrolizumab or a placebo in combination with trastuzumab and chemotherapy (5FU and cisplatin or capecitabine and oxaliplatin). The combination showed an overall response rate (ORR) of 74.4% in the pembrolizumab group compared to 51.9% in the trastuzumab-and-chemotherapy group, for a greater than 22% improvement in the ORR (95% CI 11.2–33.7; *p* = 0.00006) [71]. Progression-free survival (PFS) and OS data were not reported yet [71]. However, given the improvement in the ORR, the combination of pembrolizumab with trastuzumab and chemotherapy received accelerated FDA approval in 2021 as a first-line treatment [114]. 

Other HER2-targeted treatments have also been evaluated as second-line therapy in advanced gastric cancers. The phase II/III randomized **GATSBY trial** did not show improved OS with the HER2 antibody–drug conjugate T-DM1 when compared to taxanes in patients with HER2-positive locally advanced or metastatic gastric or GEJ cancer who had progressed on first-line therapy [66]. Median OS was 7.9 months in the T-DM1 group versus 8.6 months in the taxane group (HR: 1.15, 95% CI 0.87–1.51; one-sided *p* = 0.86) and median PFS was 2.7 months in the T-DM1 group vs. 2.9 months in the taxane group (HR 1.13, 95% CI 0.89–1.43; two-sided *p* = 0.31) [66]. 

Fam-trastuzumab-deruxtecan-nxki (T-DXd) was evaluated in the phase II **DESTINY-Gastric 01 trial**, where it was compared with chemotherapy (irinotecan or paclitaxel) in HER2-positive advanced gastric cancer patients previously treated with at least two lines of therapy, including trastuzumab [70]. The T-DXd group showed a significantly higher ORR (primary endpoint), as well as longer OS and PFS compared to the chemotherapy group (ORR 51% vs. 14% (*p* < 0.001), median OS 12.5 months vs. 8.4 months (HR 0.59; 95% CI, 0.39–0.88; *p* = 0.01), and a median PFS of 5.6 months vs. 3.5 months (HR 0.47; 95% CI, 0.31–0.71)) [70]. Based on these results, T-DXd was FDA-approved in 2022 for patients with HER2-positive gastric or GEJ adenocarcinomas who had received prior trastuzumab [115]. It was suggested that a more effective linker and payload system of T-DXd may explain the better outcomes compared to T-DM1 [70]. 

In addition to the antibody–drug conjugates, HER2-directed TKIs have also been investigated in advanced gastric or GEJ cancers, but have not demonstrated proven efficacy yet. In the phase III randomized **TyTAN trial**, lapatinib in combination with paclitaxel versus paclitaxel alone in previously treated metastatic HER2-positive patients demonstrated no significant OS and PFS benefit; 11.0 months in the lapatinib arm vs. 8.9 months in the paclitaxel arm (HR: 0.84; 95% CI 0.64–1.11; *p* = 0.1044) and 5.4 months vs. 4.4 months (HR: 0.85; 95% CI 0.63–1.13; *p* = 0.2441), respectively [67]. Lapatinib in combination with chemotherapy (capecitabine plus oxaliplatin) was again evaluated compared to chemotherapy alone in previously untreated locally advanced or metastatic HER2-positive gastroesophageal adenocarcinomas in a randomized phase III **TRIO-013/LOGiC trial** across different continents [68]. The addition of lapatinib again did not meet the primary endpoint of OS (12.2 months in lapatinib-containing arm vs. 10.5 months in chemotherapy-alone arm, HR: 0.91; 95% CI 0.73–1.12, *p* = 0.3492) [68].

### 9.2. Colorectal Cancer

HER2 amplification or overexpression is observed in about 2–6% of patients with advanced or metastatic colorectal cancer (CRC), but its prevalence is higher in RAS/BRAF-wild-type tumors with 5–14% [73,77,116]. HER2 positivity is not a prognostic marker but it is a predictive marker for HER2-targeted therapy in CRC [116]. Currently, four different regimens including T-DXd monotherapy, trastuzumab in combination with pertuzumab, or tucatinib are recommended as subsequent treatments for HER2-amplified metastatic CRC, but only tucatinib has been FDA-approved [117]. 

Analysis of a CRC cohort in the phase IIa **MyPathway basket trial** reported that dual HER2-directed therapy with pertuzumab and trastuzumab in heavily pretreated metastatic CRC patients with HER2 amplification, overexpression, and/or activating mutations showed promising results, with an ORR of 32% (95% CI 20–45), and updated results in 2021 showed an ORR of 26.2% (95% CI 17.2–36.9) [118,119]. The trastuzumab-and-pertuzumab combination in another phase II basket trial TAPUR also demonstrated efficacy in metastatic CRC patients with ERBB2 amplifications, with a disease control rate (DCR) of 54%, an ORR of 25%, median PFS of 17.2 weeks and median OS of 60.0 weeks. The study also included another 10 patients with ERBB 2/3-activating mutations, but it showed a 0% ORR [75]. The phase II Japanese **TRIUMH** study also demonstrated the promising anti-tumor activity of pertuzumab plus trastuzumab in treatment-refractory metastatic CRC patients with ERBB2 amplification (IHC3+ or FISH-positive) in tissue and/or circulating tumor DNA (ORRs 30% and 28% in patients with HER2 amplification confirmed by tumor tissue and ctDNA, respectively) [120]. 

The phase II **HERACLES-B** trial evaluated the combination of pertuzumab with T-DM1 in RAS/BRAF wild-type HER2-positive metastatic CRCs. All 31 patients were FISH-positive, with 80% having IHC 3+. Even though it did not reach its primary endpoint (ORR 9.7%, 95% CI 0–28), DCR was achieved in 77.4% of patients, with a PFS of 4.1 months (95% CI: 3.6–5.9), thus making it an option for patients with a low tumor burden who do not require significant tumor shrinkage [74].

The phase II **DESTINY-CRC01** evaluated T-DXd in previously treated HER2-positive RAS wild-type metastatic CRC. Patients were divided into three cohorts, with cohort A including HER2 IHC 3+ or IHC2+/FISH-positive patients, while cohort B and C included IHC2+/FISH-negative and IHC 1+, respectively [76,121]. The trial reported updated results in 2022, with a confirmed ORR of 45.3% (95% CI, 31.6–59.6), duration of response (DOR) of 7.0 months (95% CI, 5.8–9.5), median PFS of 6.9 months (95% CI, 4.1–8.7) and median OS of 15.5 months (95% CI, 8.8–20.8) in cohort A, making T-DXd a potential treatment for HER2-positive tumors refractory to standard treatment; however, it is not yet FDA-approved [76,121]. Cohort B and C demonstrated no responses and Cohort B showed PFS of 2.1 months and OS of 7.3 months, while cohort C showed PFS of 1.4 months and OS of 7.7 months [76]. Another phase II **DESTINY-CRC02** is currently evaluating the safety and efficacy of T-DXd in HER2-positive, RAS wild-type or -mutant advanced or metastatic CRC at both low and high doses (5.4 mg/kg and 6.4 mg/kg doses), and results are pending [122]. 

HER2-targeted TKIs were also evaluated in metastatic CRC. Lapatinib in combination with trastuzumab in the phase II **HERACLES** trial showed an ORR of 30% (95% CI 14–50), with PFS of 21 weeks and OS of 46 weeks in previously treated (median 3 prior lines) HER2-positive KRAS wild-type metastatic CRC [72]. On the other hand, tucatinib was evaluated in combination with trastuzumab in patients with chemotherapy-refractory, HER2-positive, RAS wild-type unresectable or metastatic CRC in the phase II **MOUNTAINEER trial** [77]. Initially the study was a single cohort with tucatinib and trastuzumab (cohort A, with 45 patients), but was later expanded to include the additional patients that were randomized in a 4:3 ratio to continue treatment with dual tucatinib and trastuzumab (cohort B, with 41 patients) or tucatinib monotherapy (cohort C, with 31 patients) [77]. The primary endpoint was the ORR. Combination therapy in cohorts A and B demonstrated meaningful efficacy, with an ORR of 38.1%, PFS of 8.2 months and OS of 24.1 months, while tucatinib monotherapy (cohort C) had an ORR of only 3.3% [16]. A total of 60% of patients from tucatinib monotherapy crossed over to the combination group and received an improved ORR of 17.9%, with PFS of 21.1 months [77]. Based on these results, tucatinib with trastuzumab was FDA-approved for patients with RAS wild-type, HER2-positive unresectable or metastatic CRC that progressed on fluoropyrimidine-, oxaliplatin-, and irinotecan-based chemotherapy in 2023 [123]. Given the promising results, tucatinib plus trastuzumab with modified FOLFOX6 is being further investigated compared to the standard of care in treatment-naive HER2-positive metastatic CRC in a randomized phase III MOUNTAINEER-03 trial (NCT05253651) [77].

In the above-mentioned TRIUMH trial, investigators evaluated the application of an artificial intelligence (AI)-powered HER2 quantification continuous score (QCS) in 30 tumor samples with proven HER2 amplification by HER2 FISH or ctDNA analysis, and the results were recently presented at ASCO 2023 [124]. AI powered whole-side image analyzers detect the HER2 staining intensity (negative, 1+, 2+, or 3+). The ORR in a subgroup of HER2 IHC 3+ assessed by pathologists was 34.8%, while the ORR was 42.1% in a subgroup with a HER2 3+ QCS ≥ 50% [124]. There was also improved OS in the HER2 3+ QCS ≥ 50% group compared to the <50% group (4.1 vs. 1.4 months, HR 0.12, 95% CI 0.04–0.38, *p* = 0.0000994) and an improved median OS of 16.5 vs. 4.1 months (HR 0.13, 95% CI 0.05–0.38, *p* = 0.000117), thus suggesting that HER2 QCS could provide supplemental information to gauge the precise prediction of response to anti-HER2 therapy [124].

### 9.3. Non-Small-Cell Lung Cancer 

HER2 gene-activating mutations occur in 2–3% of lung cancers [78,125]. Given that HER2 IHC 3+ or HER2 amplification by FISH is very rare in lung cancers, clinical trials evaluating the activity of trastuzumab in lung cancers with lower levels of HER2 IHC positivity have been negative [126,127]. HER2 IHC is not the ideal biomarker to use in lung cancers, but HER2-activating mutations have shown promising results as a therapeutic target [125,128]. 

Analysis of the lung cohort from a **phase II basket trial of T-DM1** showed an ORR of 44% in heavily-treated HER2-mutant lung cancers, meeting its primary endpoint ORR, with a median PFS of 5 months. Responses were observed in tumors with HER2 exon 20 insertions and point mutations in the kinase, transmembrane and extracellular domains, identified by next-generation sequencing (NGS) and low HER2 expressions identified by mass spectrometry [125]. Concurrent HER2 amplification was observed in only 11% of those responders [125]. While T-DM1 is not FDA approved for HER2-mutant NSCLC, it is otherwise recommended in previously treated HER2-mutant lung cancers based on these study results [129].

In a phase II **DESTINY-Lung01 trial** of HER2-mutant NSCLC refractory to standard treatment, T-DXd (6.4 mg/kg) demonstrated an ORR of 55% with a DOR of 9.3 months, PFS of 8.2 months and OS of 17.8 months [78]. This trial also confirmed that HER2 expression or HER2 amplification do not correlate with responses, and instead responses were observed across different HER2 mutation subtypes [78]. A total of 86% of patients had HER2 exon 20 insertions while others had single-nucleotide variants in exon 19 or 20 of the kinase domain or in exon 8 of the extracellular domain. A follow-up phase II **DESTINY-Lung02 trial** evaluated T-DXd 5.4 mg/kg every 3 weeks compared to 6.4 mg/kg every 3 weeks in previously treated HER2-mutant metastatic NSCLC [130]. Analysis showed an ORR of 49% and 56% in the 5.6 mg/kg group and the 6.4 mg/kg group, respectively, confirming that T-DXd demonstrated meaningful clinical response at both doses, with a safety profile favoring the T-DXd 5.4 mg/kg dose [130]. T-DXd was FDA-approved as a second-line treatment for unresectable or metastatic NSCLC patients with activating HER2 mutations on 11 August 2022, based on those results [131]. 

### 9.4. Ovarian Cancer

HER2 overexpression has been observed in an average of 11–16% of ovarian tumor samples in recent studies, but could be observed in a wide range of variation from 8% to 66% [79,132]. In a multicenter GINECO study of 320 patients, HER2-gene overexpression and amplification was found in 6.6% of tumors [132]. The study also analyzed that there was no association between HER2 status and other prognostic factors including the tumor histology, grade, and stage, as well as ascites, the debulking status and age [132]. In addition, HER2 status did not predict prognosis in terms of OS and PFS [132]. The efficacy of single-agent trastuzumab was found to be low, with a 7.3% ORR, DOR of 8 weeks (range 2–104 weeks), and PFS of only 2.0 months in heavily treated patients with persistent or recurrent HER2-overexpressed epithelial ovarian or primary peritoneal carcinoma [79]. The ORR was slightly higher with T-DM1 in a phase II basket trial of an ovarian cancer cohort, with an ORR of 17% in previously treated HER2-positive ovarian cancers [133]. Currently, anti-HER2 therapy is not yet approved or recommended in HER2-positive ovarian cancers. 

### 9.5. Endometrial Cancer

HER2 amplification is found in about 12% of endometrial cancers, and HER2 overexpression by IHC staining was present in 44% of tumors analyzed in the GOG 177 trial, with the highest amplification in high-grade serous and clear-cell cancers [134,135]. HER2 gene amplification is found in 17–30% of high-grade endometrial cancers, while HER2 overexpression can be found in up to 80% of cases [136]. Some studies have shown association between HER2 positivity with tumors of a higher grade, stage and lymph node positivity, and worse OS, while other studies have showed that a positive HER2 status has no correlation with OS [136,137,138]. Moreover, the clinical application of HER2 overexpression or gene amplification is controversial in endometrial cancer. Anti-HER2 therapy in advanced/recurrent endometrial cancer has demonstrated poor response rates. Trastuzumab as a single agent did not demonstrate activity against advanced or recurrent endometrial carcinomas with HER2 overexpression or HER2 amplification in a phase II trial, with an ORR of 0%, PFS of 1.84 months and OS of 7.85 months [134]. In addition, the activity of combined trastuzumab and pertuzumab was also low in a cohort of endometrial cancer in the phase II MyPathway basket trial in HER2-positive tumors, with a 4.3% ORR [118,119]. In addition, evaluation of the endometrial cohort in the phase II TAPUR trial reported only a mild anti-tumor activity of combined trastuzumab and pertuzumab in heavily treated patients with HER2 amplifications, with an ORR of 7%, DCR of 37%, PFS of 16 weeks and OS of 61 weeks [82]. 

On the other hand, in a phase II randomized trial of HER2-positive advanced or recurrent uterine serous carcinomas, patients were randomized into a trastuzumab-plus-carboplatin-and-paclitaxel (T arm) or carboplatin-and-paclitaxel group (C arm) [80]. The median PFS in the overall population was 8.0 months (C arm) versus 12.6 months (T arm) (HR: 0.44; 90% CI 0.26–0.76; *p* = 0.005) [80]. The median PFS was even higher in patients receiving primary treatment than with recurrent disease. The median PFS was 9.3 months (C arm) vs. 17.9 months (T arm) (HR: 0.4, 90% CI 0.2–0.8; *p* = 0.013) in 41 patients undergoing primary treatment, while PFS was 6 months (C arm) versus 9.2 months (T arm) (HR: 0.14, 90% CI 0.04–0.53; *p* = 0.003) in 17 patients with recurrent disease [80]. An updated analysis in 2020 supported this benefit, with PFS of 8.0 months (C arm) vs. 12.9 months (T arm), and OS of 24.4 months vs. 29.6 months, respectively (HR 0.58; 90% CI 0.34–0.99; *p* = 0.046) [81]. Based on these results, the combination of trastuzumab with carboplatin and paclitaxel is recommend for HER2-positive uterine serous carcinoma patients with stage III/IV disease as a primary therapy or for recurrent disease, whereas it is recommended as a category 2 option for HER2-positive carcinosarcoma in both disease settings [139]. 

T-DM1 showed only minimal activity in HER2-positive endometrial cancers in a phase II basket trial, with an ORR of 22% [133,140]. T-DXd is currently being evaluated in another basket trial of HER2-positive tumors, **DESTINY-PanTumor02**, which includes endometrial carcinoma (NCT04482309). 

### 9.6. Urothelial Cancer

Urothelial cancer has the third highest rate of HER2 overexpression after breast and gastric cancers, with a wide range from 8.5% to 81% of cases and mutations in and amplifications of HER2 in approximately 6–17% of tumors [141,142]. HER2 overexpression could be correlated with lymph node positivity, a greater tumor stage, and lower OS, and, thus, overall a worse prognosis [142,143,144]. However, HER2 amplification could also be a bystander event in most muscle-invasive bladder cancers (MIBCs). Given that it is characterized by a high rate of genomic alterations, this individual alteration may not be a significant oncogenic driver [110]. A study by Kiss et al. found that HER2 overexpression in MIBC does not always correlate with HER2 amplification [110]. It showed that HER2 amplification does not always lead to HER2 overexpression, given that HER2 expression was negative in 5 out of the 16 samples with HER2 amplification, while HER2 overexpression could also be seen in samples without HER2 amplification, suggesting that HER2 overexpression is not solely driven by HER2 amplification [110]. Thus, the study recommended the evaluation of HER2 at all three levels of the DNA, RNA and protein, to accurately characterize HER2 alterations for HER2-targeted trials [110]. Regardless, HER2 positivity is one of the significant prognostic factors for recurrence-free survival in non-muscle-invasive bladder cancer (NMIBC), and HER2 has been investigated as a therapeutic target in HER2-positive urothelial cancers [145]. 

A phase II trial of trastuzumab, paclitaxel, carboplatin, and gemcitabine in HER2-positive metastatic urothelial cancers, as a first-line treatment, showed an ORR of 70%, with a DOR of 7.1 months, a PFS of 9.3 months and an OS of 14.1 months, demonstrating a benefit of anti-HER2 therapy [141]. However, 22.7% experienced grade 1 to 3 cardiac toxicity [141]. The urothelial cancer cohort from the phase II **MyPathway basket trial** showed that trastuzumab and pertuzumab achieved an ORR of 18.2% in HER2-positive metastatic urothelial cancer patients, which was lower than the ORR reported in HER2-positive salivary gland cancer (61.6%), CRC (26.2%) biliary tract cancer (BTC) (22.5%) and non-small-cell lung cancer (NSCLC) (25.9%), but higher than in pancreatic cancer (10%), ovarian cancer (8.3%) and uterine cancer (4.3%) [118]. 

Anti-HER2 TKIs and antibody–drug conjugates are currently being investigated in different clinical trials [142]. Currently, anti-HER2 therapy is not yet approved or recommended in bladder cancer.

### 9.7. Salivary Gland Tumor

Out of the head and neck cancers, HER2 overexpression or amplification can be found in 30–40% of salivary gland tumors, and anti-HER2 therapy has proven to have benefits [146,147]. An initial case study suggested the promising anti-tumor activity of trastuzumab in previously treated salivary gland tumors [148]. A phase II Japanese study of trastuzumab with docetaxel in previously treated advanced salivary duct carcinoma showed impressive results, with an ORR of 70.2% and a clinical benefit rate of 84.2%, with a PFS of 8.9 months and OS of 39.7 months [146]. The combination of trastuzumab and docetaxel showed similar results in another phase II trial of unresectable, recurrent or metastatic HER2-positive salivary duct carcinoma (0–3 prior lines of treatment), with an ORR of 69.8%, PFS of 7.9 months, and OS of 23.3 months [149]. 

The salivary gland cancer cohort from the phase II **MyPathway basket trial** showed that combined trastuzumab and pertuzumab has an ORR of 60%, with a DOR of 9.2 months, PFS of 8.6 months and OS of 20.4 months in previously treated patients (1–3 prior lines) with advanced HER2-positive salivary gland tumors [83]. In addition, T-DM1 showed partial responses in two out of the three patients with metastatic HER2-positive salivary gland tumors in the phase II NCI-**MATCH basket (EAY131) trial** [150]. T-DM1 was also evaluated in another phase II basket trial in HER2-positive metastatic salivary gland tumors, with an ORR of 90%, including five complete responses in patients previously treated with trastuzumab, pertuzumab and anti-androgen therapy [151]. T-DXd was associated with an ORR of 47%, a DOR of 12.9 months and a PFS of 14.1 months in advanced HER2-positive salivary duct cancers in an analysis of two phase I studies [152]. Trastuzumab monotherapy, trastuzumab and pertuzumab, trastuzumab with docetaxel, and T-DM1 are recommended for use in HER2-positive salivary gland tumors, and T-DXd can be considered as an option [153].

### 9.8. Biliary Tract Cancer

HER2 overexpression or amplification is present in about 17% of extrahepatic cholangiocarcinomas, and 5% of intrahepatic cholangiocarcinomas, making them a promising target for the treatment of advanced BTC [84]. The phase IIa **MyPathway basket trial** of trastuzumab with pertuzumab in 39 patients with previously treated HER2-amplified and/or -overexpressed metastatic BTC showed an ORR of 23%, with DOR of 10.8 months, PFS of 4 months and OS of 10.9 months [84]. On the other hand, the phase II multi-histology basket trial of T-DM1 showed low activity, with an ORR of 12% in BTC [133,140]. T-DXd was also evaluated in the phase II Japanese **HERB trial,** with an ORR of 36.4%, PFS of 4.4 months and OS of 7.1 months in 22 patients with HER2-positive BTC refractory to a gemcitabine-based regimen [154]. Even though there is no currently FDA-approved anti-HER2 therapy for BTC in the setting of limited data, the combination of trastuzumab and pertuzumab is recommended in certain circumstances as a subsequent line of treatment for advanced HER2-positive tumors [155]. 

Table 3 summarizes all current FDA-approved anti-HER2 targeted agents across caners including breast.

## 10. Mechanisms of Resistance to ERBB Inhibitors

Over the past three decades, there have been major advances and innovative discoveries in HER2-positive breast cancer, with significantly improved survival rates and an increased number of approved drugs. However, multiple mechanisms of resistance to these targeted drugs that allow an escape from HER2 inhibition also exist, which lead to disease progression [156]. Sometimes these mutations are present immediately and the cancer does not exhibit an initial response, and other times acquired resistance develops following an initial and strong clinical response. 

There are a few ways in which intrinsic mutations/alterations can occur, which then benefit the cancer cells and confer resistance to therapy. We will briefly review them.

HER2 mutations can occur in the juxtamembrane region, which contains the binding epitope for trastuzumab, leading to a truncated form of HER2 (called p95HER2). P95HER2 lacks the HER2 antibody-binding region, so it is no longer susceptible to trastuzumab. However, this truncated form of HER2 retains kinase activity and is susceptible to inhibition by TKIs [156].

HER2 (L755S) is the most common alteration associated with anti-HER2 trastuzumab resistance. Unfortunately, this mutation has also developed resistance to the dual blockade of trastuzumab and pertuzumab and has reduced sensitivity to T-DM1. Some second-generation TKIs have shown promising results demonstrating that they can overcome this resistance and may be a therapeutic alternative for the 3% of patients that harbor this mutation [157]. This mutation can result in hyperactivation of the MAP and PI3K/AKT/mTOR pathways and subsequent resistance to both reversible and irreversible HER2 TKIs. There is promise that anti-HER2 TKIs can work if MEK inhibitors, such as selumetinib (AZD6244), or PI3K inhibitors are added. These combinations are a new novel targeted strategy to overcome HER2(L755S) resistance with anti-HER2 treatment, which is currently being explored [158].

A splice variant that eliminates exon 16 in the extracellular domain of the HER2 receptor has also been identified; this isoform is resistant to trastuzumab. This variant does not eliminate the trastuzumab epitope on HER2, but appears to stabilize the HER2 homodimers and may potentially prevent their disruption upon binding by the antibody, leading to ineffective trastuzumab activity [156].

Point mutations or amino acid insertions in the HER2 gene have been identified in other cancers that we have discussed, including non-small-cell lung cancers (NSCLC), as well as gastric, colorectal and head and neck cancers [156]. These point mutations identified have been shown to confer resistance to lapatinib and trastuzumab [156]. In 2022, the FDA approved T-DXd for cases of previously treated metastatic NSCLC carrying a HER2-activating mutation based on the DESTINY-LUNG02 phase II trial, which tested the drug on patients with single nucleotide variants or exon 20 insertions [130].

There is also the possibility of diminishing the drug’s ability to bind to its target epitome by the co-expression of another protein that binds to the epitome instead. Mucin-4 (MUC4), a membrane-associated glycoprotein, can be overexpressed and can co-localize with HER2, effectively masking the binding site for trastuzumab. MUC4 is overexpressed in all trastuzumab-resistant cell lines [159].

Activation of compensatory pathways can occur through receptor tyrosine kinases and through intracellular kinases, which can also lead to the drug’s inability to work. Signaling through other receptor tyrosine kinases can transactivate HER2 and amplify signal transduction downstream, thus bypassing the inhibitory effect of lapatinib, neratinib, tucatinib or trastuzumab. HER2 can heterodimerize with other ErbB family members, and signaling initiated by ligands of EGFR, HER2 or HER4 can bypass the antiproliferative effects of trastuzumab. Research has shown that the increased expression of EGFR and HER3 ligands results in activation on EGFR and HER3, as well as increased EGFR/HER2 heterodimers in trastuzumab-resistant cells. This is consistent with data showing that trastuzumab is unable to block the ligand-induced heterodimerization of HER2. Another mechanism, which increases the availability of ErbB ligands, is a mutation that leads to the activation of TGFb receptors. TGFb promotes ligands that further activate HER3, and, therefore, there is more HER3 available to heterodimerize with HER2 [156]. 

The gain of function mutations in PIK3CA/AKT pathways, which are downstream of the HER2 receptor and are caused by a loss of the tumor suppressor PTEN, can confer resistance to HER2 inhibitors. In fact, PIK3CA/AKT is the most frequent tumor somatic alteration in breast cancer, occurring in >30% of invasive tumors [156]. There are also studies showing that neither PIK3CA mutations or low PTEN is associated with poor prognosis, aside from treatment resistance [159]. While PI3K inhibitors are known to be effective in HR-positive, HER2-negative breast cancer, with alpelisib currently FDA-approved in this setting, these drugs are not currently used for HER2-positive breast cancer. There are some preclinical data proving the activity of PI3K inhibitors in combination with trastuzumab in HER2-positive tumor cell lines in a mouse model [160].

SRC is a proto-oncogene, and a non-receptor tyrosine kinase that functions in the regulation of cell growth. Increased SRC activation and subsequent phosphorylation has been shown to be associated with trastuzumab resistance [159]. There is ongoing research in which targeting SRC sensitizes cells to trastuzumab. The most studied SRC inhibitor is dasatinib. A small study of 29 patients with metastatic HER2-positive breast cancer showed that the combination of dasatinib, trastuzumab and paclitaxel shows strong efficacy, achieving a median PFS of 23.9 months [161]. However, this study was limited due to its small number of participants, and this remains an ongoing area of research.

Cell-cycle regulators have also shown to play an important part of HER2-driven tumorigenesis. There is a high correlation between a high copy amount of the CCND1 gene, which encodes Cyclin D1, and trastuzumab resistance [159]. Cyclin-dependent kinase 4 and 6 (CDK4/6) inhibitors, such as abemaciclib, can be active in trastuzumab-resistant, HR-positive, HER2-positive tumors. In the MonarcHER phase II trial, patients with HR-positive, HER2-positive advanced breast cancer previously treated with at least two lines of anti-HER2 therapy were treated with a combination of abemaciclib with trastuzumab and fulvestrant, a selective estrogen receptor degrader (group A), abemaciclib and trastuzumab (group B), or the standard of care with trastuzumab and the physician’s choice of chemotherapy (group C). Group A achieved a median PFS of 8.3 months, compared to 5.6 months in group B and 5.7 months in group C [162]. This trial proved that abemaciclib can provide a chemotherapy-free treatment option in HR+/HER2+ patients who have already been pretreated with trastuzumab. Another mechanism of resistance is though the upregulation of cyclin E1, which is encoded by the CCNE1 gene and which regulates cyclin-dependent kinase 2 (CDK2). It has been found that patients who have been heavily pretreated with trastuzumab can have cyclin E1 amplification/overexpression, which in turn leads to the resistance of trastuzumab [159].

There are host immunity factors that are required for trastuzumab to actually work. Trastuzumab exerts some of its anti-tumor effects via the engagement of host immune effectors, such as antigen-specific T cells and antibody-dependent, cell-mediated toxicity (ADCC). Thus, problems with host factors that affect this immunomodulatory function can also contribute to trastuzumab resistance. An inability of host cells to mediate a sufficiently strong ADCC response may contribute to resistance [159].

Sometimes, resistance can be overcome with combinations of different anti-HER2 therapies. There has also been evidence showing that the combination of targeted therapy early in the disease course can prevent drug resistance. Some approaches in recent years include novel tyrosine kinases with the subsequent inhibition of intracellular signaling (PI3K/AKT/mTOR and CDK4/6 inhibitors), as well as other newer approaches that are in the design process to attack other tumor weaknesses. These approaches include categories like immunotherapy and autophagy blockade [163]. 

## 11. Mechanism and Management of the Most Relevant Toxicities

There are a few common or major adverse events that can be caused by anti-HER2 therapies.

### 11.1. Cardiotoxicity

Cardiotoxicity with HER2-directed therapy can lead to a decreased left ventricular ejection fraction (LVEF). There is an even higher risk for those patients treated in combination with HER2-directed therapy and anthracyclines versus chemotherapy alone. When cardiotoxicity is related to anthracyclines, it is usually dose-dependent; however, treatment with anti-HER2 therapy is not associated with cumulative doses and is usually generally reversible.

A reduced LVEF with anti-HER2 monoclonal antibodies can range from 2–6%, depending on the combination of chemotherapy that is used (anthracycline or not) [101]. Reports of cardiotoxicity with antibody–drug conjugates have been low (<1% to 2%) [95], and patients seem to recover after therapy interruption. There have never been any reported events of cardiac failure associated with an LVEF decrease in the studies conducted that have assessed the safety of the antibody drug conjugates. There are also even lower rates of cardiac toxicity with TKIs compared to standard cardiotoxicity with trastuzumab or pertuzumab [163]. 

Baseline LVEF evaluation before the initiation of anti-HER2 therapy is necessary. Guidelines on the cardiovascular monitoring of patients receiving HER2-directed therapy have mostly been based on studies with patients receiving trastuzumab. If patients experience cardiotoxicity, they may be completely asymptomatic. For this reason, cardiac monitoring should occur every 3 months during treatment and every 6 months following completion of therapy for up to 2 years, regardless of whether or not they received anthracyclines [163]. During the initial evaluation of patients, cardiac risk factors should be optimized with the encouragement of a balanced and healthy lifestyle, and the possible implementation of appropriate cardiac medications, with the assistance of a cardiologist, if they already have baseline LV dysfunction [163]. During treatment monitoring, if their LVEF does decrease, cardiac medications should be initiated, and HER2-directed therapy should be discontinued. Heart function should then be reassessed in 4 weeks; if their LVEF improves again and the patient feels back to baseline, the patient can be re-challenged with HER2-directed therapy and the monitoring continues [164].

### 11.2. Gastrointestinal Toxicity

GI toxicity, specifically diarrhea, often occurs with tyrosine kinase inhibitors. Diarrhea is most frequent with neratinib and pyrotinib compared to other TKIs. The use of standard anti-diarrheal medications like loperamide may alleviate symptoms. Based on the CONTROL trial, there was significantly improved tolerability of neratinib if management was pro-active and preventative at the start of therapy. Great effectiveness was shown with scheduled loperamide with or without colestipol, and also with a simple dose escalation of neratinib [165]. Diarrhea was also frequently experienced in patients who received pertuzumab. The diarrhea mostly occurred during the first treatment cycle, and again it also responded well to standard anti-diarrheal agents [166].

### 11.3. Skin Rash 

Rashes on the skin can also can occur with TKIs, especially those that can cross-react with the EGFR receptor. It can also occur with pertuzumab, but is generally less common compared to skin rash with TKI. The skin rashes are usually papulopustular or acneiform lesions. To prevent skin rashes, sunblock is recommended that is at least SPF 15, as well as frequent moisturization. If a rash does appear and is grade 2–3, tetracyclines including doxycycline or minocycline can be used, as well as the implementation of topical corticosteroids [166]. Hand–foot syndrome is also a skin reaction that can be observed in patients receiving tucatinib. If a TKI has higher specificity towards HER2 over EGFR, it may have less GI and skin effects overall (an example of this is tucatinib), and it is generally better tolerated [166].

### 11.4. Liver Toxicity 

Toxicity to the liver can occur, and a patient may present with elevated liver enzymes from agents like T-DM1 and tucatinib. If this occurs, the medication may need to be held and dose adjustments may need to be made moving forward [166]. TDM-1 specifically has been known to cause a transaminitis or bilirubin rise in up to 20–80% of patients. In at least 5% of patients, it has even been reported that enzyme levels rose to above five times the upper limit of normal. Instances of acute liver injury and death from hepatic failure have also been reported from TDM-1. It is believed that some of these cases can be attributed to either direct liver injury or acute sinusoidal obstruction syndrome. Of recent, there have been cases reported of non-cirrhotic portal hypertension in patients who are long-term users of trastuzumab emtansine [167]. To help mitigate liver toxicity, it is also important to see if the patient is taking any other mediations that moderately or significantly inhibit CYP3A, and this can be discussed with pharmacy if assistance is needed. 

### 11.5. Thrombocytopenia

Low platelets can occur with T-DM1. The chemotherapy molecule DM1 can lead to the impairment of megakaryocyte differentiation. It is the most common toxicity leading to discontinuation of T-DM1. Dose reduction is the recommended strategy. 

### 11.6. Interstitial lung (ILD)

ILD disease has been attributed to T-DXd and was first reported in 13.6% of patients in the DESTINY-Breast01 trial [157]. The risk was higher if the patients were >65 years of age, or had a baseline O_2_ saturation of <95%, acute kidney injury, or concomitant respiratory comorbidities. If a patient shows any evidence of lung symptoms, therapy needs to be promptly interrupted and the patient needs treatment with steroids [166].

### 11.7. Peripheral Neuropathy

Neuropathy is also a side effect that should be a consideration in patients about to receive T-DM1. This is due to the fact that emtansine is a microtubule inhibitor. This is especially important because peripheral neuropathy is a significant toxicity of taxanes, which patients have often already been exposed to as part of their previous therapy. However, T-DM1 is associated with a relatively lower risk of peripheral neuropathy than a taxane-based regimen. Nevertheless, peripheral neuropathy should be a consideration when selecting therapy for HER2-positive breast cancer patients at high risk of developing it, or in those patients who already have it as a pre-existing condition prior to starting treatment [168].

## 12. Look to the Future: Novel Anti-HER2 Therapy in Development

Over the years, significant advancements have been made in the development of HER2-targeted therapies, including monoclonal antibodies, small-molecule inhibitors, and immune checkpoint inhibitors. These emerging therapies have garnered attention for their potential to further improve outcomes for patients with cancers that express HER2. 

### 12.1. Antibody–Drug Conjugates 

Trastuzumab duocarmazine (SYD985) is an ADC comprising trastuzumab covalently bound to duocarmycin, an alkylating agent, via a cleavable linker. Based on the results of a phase I trial, in which SYD985 showed significant clinical activity in heavily pretreated patients with HER2-positive and HER2-low breast cancer, the phase III **TULIP trial** evaluated SYD985 versus the physician’s choice of chemotherapy in 431 HER2-positive breast cancer patients who progressed after two or more HER2-targeted therapies [169]. SYD985 was associated with a significant improvement in PFS (7.0 vs. 4.9 months; HR 0.64, 95%CI 0.49–0.84; *p* = 0.002). The most common treatment-related adverse event was ocular toxicity [169,170]. The drug is currently under evaluation by the FDA.

ARX788 is a homogeneous, site-specific, and highly stable ADC that comprising an anti-HER2 antibody and amberstatin (AS269), a tubulin inhibitor. Preclinical studies with ARX788 demonstrated promising activity in HER2-positive, HER2-low, and T-DM1-resistant tumors [171]. In two phase 1 studies in HER2-positive breast cancers (ACE-Breast-01) and solid tumors (ACE-Pan tumor-01), ARX788 showed an ORR of 74% (14/19) and 67% (2/3), respectively [172,173]. The most common grade 3–4 adverse events (AE’s) were ocular toxicity (5.7%) and pneumonitis (4.3%) in the ACE-Breast-01 trial, and pneumonitis (2.9%) and fatigue (2.9%) in the ACE-Pan tumor-01 trial. In another phase I study evaluating only patients with HER2-positive metastatic breast cancers, the ORR was 65.5% (19/29) and the median PFS was 17.0 months [174]. ARX788 is currently being studied in several phase II trials (NCT04829604, NCT01042379, and NCT05018702). In addition, the phase II-III trial (NCT05426486) is comparing ARX788 combined with pyrotinib maleate versus TCHP (trastuzumab, pertuzumab, docetaxel and carboplatin) as neoadjuvant treatment in patients with stage II-III HER2-positive breast cancer. In recent news, the phase III ACE-Breast-02 trial was reported to have met its pre-specified interim primary efficacy endpoint, demonstrating improved PFS with ARX788 compared to lapatinib with capecitabine in previously treated patients with HER2-positive locally advanced or metastatic breast cancers; the formal results are still pending [175].

Disitamab vedotin (RC48) combines the humanized anti-HER2 antibody hertuzumab with monomethyl auristatin E (MMAE), a tubuline-targeting agent, via a cleavable linker. Compared with trastuzumab, hertuzumab exhibits greater affinity for HER2 and produces more potent cytotoxicity [176]. Multiple ongoing clinical trials are evaluating disitamab vedotin in HER2-positive solid tumors, including urothelial bladder cacinomas (NCT05495724), advanced or metastatic colorectal cancer (NCT05493683, NCT05333809), and locally advanced or metastatic NSCLC (NCT05847764). There have also been clinical trials evaluating the use of disitamab vedotin in breast cancers, two of which were evaluated in a pooled analysis, showing efficacy in HER2-positive- and HER2-low-expressing subgroups [177]. In the HER2-positive subgroup, the ORR and median PFS were 40.0% and 6.3 months, respectively. In the HER2-low-expressing subgroup, the ORR and median PFS were 39.6% and 5.7 months, respectively. The most common grade ≥3 TRAEs were neutropenia (16.9%), elevated gamma-glutamyl transpeptidase (12.7%), and fatigue (11.9%). Disitamab vedotin is also currently being studied in combination with penpulimab as neoadjuvant therapy in HER2-positive breast cancers (NCT05726175). 

### 12.2. Tyrosine Kinase Inhibitors

Epertinib (S-222611) is a reversible inhibitor of HER2, EGFR and HER4, that demonstrated more potent effect compared to lapatinib, as well as antitumor activity in a brain metastasis model of HER-positive breast cancer in preclinical studies [178,179]. Several trials have evaluated treatment with epertinib in solid cancers, including a phase I/II study of epertinib plus trastuzumab with or without chemotherapy in patients with pre-treated HER2-positive metastatic breast cancer, with or without brain metastases. The ORR for epertinib combined with trastuzumab was 67% (6/9), and with trastuzumab plus capecitabine it was 56% (5/9). A partial response (PR) was achieved in two of three patients with brain metastases and one of two patients, respectively [180].

DZD1516 is a reversible and selective HER2 kinase inhibitor that has blood–brain barrier penetration. A phase I study by Zhang et al. evaluated 23 patients with heavily pre-treated HER2-positive metastatic breast cancer; among them, 15 patients (65.2%) had CNS metastases at the study onset [181]. With a median of seven lines of therapy of prior treatment, the best antitumor efficacy in the intracranial, extracranial, and overall lesions was stable disease (6 of 23 patients). DZD1516 was well tolerated, with the majority of AEs being grade 1 and reversible. The most common AEs (of any grades) were headache, vomiting, and anemia [182]. 

JBJ-08–178-01 is a new mutant-selective HER2 kinase inhibitor that reduces both the kinase activity and protein levels of HER2 in lung cancer. In preclinical models of HER2-mutant cancers, JBJ-08–178-01 demonstrated a dose-dependent inhibition of HER2 [183]. JBJ-08-178-01 also exhibits more selectivity for HER2 mutations over wild-type EGFR compared with other EGFR/HER2 TKIs. High doses of JBJ-08–178–01 (100 mg/kg/day or 50 mg/kg BID) achieved comparable levels of inhibition to neratinib (40 mg/kg/day) and tucatinib (100 mg/kg BID). JBJ-08–178-01’s ability to reduce both the kinase activity and protein levels of HER2 represents a combination of mechanisms that may lead to better efficacy and tolerance in patients with NSCLC carrying HER2 mutations or amplification [182,184]. 

### 12.3. CAR T-Cell Therapy

Chimeric antigen receptor (CAR) T-cell therapy is a form of immunotherapy that involves reprogramming a patient’s own immune T-cells to better target and attack cancer cells in the body. While CAR T-cell therapy has been shown to be remarkably effective in the treatment of hematological malignancies, its use in solid tumors remains challenging due to factors such as a hostile tumor microenvironment (TME) and heterogeneous antigen expression. Strategies to overcome these challenges include using alternative immune cells, such as human macrophages with CARs (CAR-M) [185,186]. 

CT-0508 is a cellular product composed of autologous macrophages derived from peripheral blood monocytes, which is genetically modified with an adenoviral vector carrying an anti-HER2 CAR-M. The currently ongoing first-in-human, multicenter, phase 1 study of adenovirally-transduced anti-HER2 CAR-M aims to assess the safety, tolerability, and manufacturing feasibility of CT-0508 in patients with recurrent or metastatic HER2-overexpressing solid tumors [187]. Seven patients have been treated and preliminary results indicate a favorable tolerance to CT-0508, with most AEs being grade 1 or 2, although there were five serious AEs that included two cases of cytokine release syndrome (CRS), one infusion reaction, one case of gastrointestinal hemorrhage, and one case of worsening dyspnea related to progressive disease. However, there were no dose-limiting toxicities or AEs leading to discontinuation or dose modification. Regarding efficacy, the best overall response was stable disease (*n* = 3) and one patient progressed, with a median follow up of 8 weeks [188].

Another strategy in development is CAR-engineered NK cells, which can be generated from peripheral blood cells, from stem cell sources, or from NK-92 cell lines instead of using a patient’s own immune cells [189]. CAR NK-cell therapy appears to be safer than CAR T-cells because they do not induce CRS and utilize multiple mechanisms to promote cytotoxicity. CAT-179, a HER2-targeted allogeneic CAR-NK, has been evaluated in preclinical models of HER2-amplified ovarian and gastric cancer. In the first model, evaluating mice with HER2+ ovarian cancer cells, the CAT-179-dosed animals showed a 95% decline in tumor burden (*p* < 0.0001) and a significant survival benefit relative to the animals dosed with control NK cells (*p* < 0.0001). In the second model, evaluating mice with HER2+ gastric cancer cells, the CAT-179 dosed animals showed a 96% durable tumor regression and significant survival benefit relative to the animals dosed with control NK cells (*p* < 0.0001) [189]. These preclinical results demonstrate the potential of CAT-179 as a novel, durable therapy. 

### 12.4. Targeted Protein Degraders

Many proteins involved in cancer pathways are considered “undruggable”, that is, challenging as targets using traditional small-molecule inhibitors due to their structure or lack of suitable binding sites [176]. Targeted protein degradation (TPD) is an alternative strategy that has expanded the druggable proteome for cancer treatment. TPD therapies can be grouped into three main classes based on their molecular designs and mechanisms of action: immunomodulatory drugs (IMiDs), selective estrogen receptor degraders (SERDs), and proteolysis-targeting chimeras (PROTACs). 

PROTACs work by recruiting an E3 ubiquitin ligase to a specific target protein, resulting in the tagging of the protein with ubiquitin molecules and subsequent degradation by the cellular machinery [190]. In a proof-of-concept study, Maneiro et al. developed a specific trastuzumab-PROTAC conjugate, which would selectively bind to HER2 receptors on tumor cells and induce endosomal internalization and the release of the PROTAC, which would then selectively target and degrade the bromodomain-containing protein 4 (BRD4—an attractive oncogenic target with a role in transcriptional dysregulation) [191]. Results showed the specific degradation of BRD4 in HER2-positive cells, with no degradation observed in HER2-negative cells, demonstrating that this antibody–PROTAC strategy can target specific molecules for degradation in selected tissues by combining the catalytic potency of PROTACs with the tissue specificity of ADCs. 

### 12.5. Checkpoint Inhibitors

HER2-targeted agents combined with immune checkpoint inhibitors have shown promise in clinical trials for various types of cancer, including HER2-positive breast cancer. The phase Ib/II trial PANACEA investigated the combination of trastuzumab and pembrolizumab in patients with HER2-positive metastatic breast cancer who had been heavily pretreated [192]. In the PD-L1-positive cohort, 6 of the 40 patients (15%) achieved an objective response and the median PFS was 2.7 months. In the PD-L1-negative cohort, there were no objective responses, and the median PFS was 2.5 months. The authors concluded that pembrolizumab plus trastuzumab was safe and showed activity and durable clinical benefit in patients with PD-L1-positive, trastuzumab-resistant, HER2-positive metastatic breast cancer. 

There is also a lot of interest in combinations of HER2-directed ADCs and immunotherapy. The phase II KATE2 trial compared T-DM1 plus atezolizumab versus a placebo in patients with HER2-positive locally advanced or metastatic breast cancer who received prior trastuzumab- and taxane-based therapy [193]. T-DM1 plus atezolizumab did not demonstrate a clinically significant PFS advantage compared with T-DM1 plus placebo, with a median PFS of 8.2 months versus 6.8 months (stratified HR 0.82, 95% CI 0.5–1.23; *p* = 0.33), respectively. In the PD-L1-positive subgroup, an objective response was achieved by 30 (54%) of the 56 patients in the atezolizumab group versus 9 (33%) of the 27 patients in the placebo group. This combination is now being investigated specifically in patients with HER2-positive and PD-L1-positive metastatic breast cancer in the ongoing phase III KATE3 trial (NCT04740918).

NKG2A is a novel immune checkpoint target; it is a receptor found on the surface of tumor-infiltrating immune cells, including NK cells and CD8+ T-cells. It interacts with its ligand, HLA-E, which is often expressed on the surface of cancer cells. This interaction sends inhibitory signals to immune cells, dampening their cytotoxic activity against the cancer cells. Monalizumab is a novel antibody that blocks the interaction between NKG2A and HLA-E and can also enhance the cytotoxic potential of trastuzumab [194]. This combination was investigated in the phase I/II MIMOSA trial in patients with previously treated HER2-positive metastatic breast cancers. While it was found to be well tolerated, results showed no objective responses (0/11); thus, the primary endpoint was not met and the study did not proceed to phase II.

### 12.6. Bispecific Antibodies

Bispecific antibody therapy is a novel approach in cancer treatment that involves the use of antibodies designed to simultaneously bind to two different antigens or two separate epitopes on the same antigen. When binding to two different antigens simultaneously, bispecific antibodies can act as bridges between different cells, such as a tumor cell and an immune cell, facilitating the immune-mediated destruction of the tumor cells. When binding to two different epitopes on the same antigen of a tumor cell, it can synergistically block signaling pathways for enhanced therapeutic efficacy, while minimizing the side effects that would otherwise result with two separate drugs. Multiple bispecific antibodies are currently in development. 

Zanidatamab (ZW25) is a humanized, bispecific, IgG1 antibody that targets the extracellular domain (ECD) IV and the dimerization domain (ECD II) of HER2, the same domains targeted by trastuzumab and pertuzumab, respectively. This bispecific targeting of two non-overlapping epitopes on HER2 can induce HER2 receptor clustering, creating a larger meshwork structure than a monoclonal antibody does. This, in turn, promotes robust internalization, lysosomal trafficking, and degradation [195]. Zanidatamab in combination with palbociclib and fulvestrant is currently being studied in an ongoing phase II trial for previously treated HR-positive/HER2-positive metastatic breast cancer (NCT04224272). Zanidatamab is also being studied in combination with evoparcept (ALX148, a CD47 blocker) in an ongoing phase Ib/II study for previously treated HER2-positive and HER2-low breast and gastroesophageal cancer (NCT05027139).

KN026 is another bispecific antibody that binds to ECD II and ECD IV on HER2. KN026 has demonstrated anti-tumor activity over cell lines with different HER2 expression levels in preclinical data [196]. Results of a phase I trial of KN026 in heavily pretreated patients with HER2-positive metastatic breast cancer showed that KN026 was well tolerated and had efficacy comparable to a trastuzumab/pertuzumab duo, even in the heavily pretreated patients, with a 28% ORR and a median PFS of 6.8 months [197]. Several ongoing trials are currently investigating KN026 in HER2-positive breast cancer (NCT04521179, NCT04881929, NCT04778982). 

Runimotamab (BTRC4017A) is an anti-HER2/anti-CD3 T-cell-dependent bispecific antibody. By binding simultaneously to CD3 receptors on T cells and HER2 receptors on tumor cells, it creates a bridge between these cells, potentiating the T-cell cytotoxic effect against HER2-positive tumor cells [198]. An ongoing phase Ia/Ib study is currently assessing runimotamab in combination with trastuzumab in pretreated patients with locally advanced or metastatic HER2-expressing cancers (NCT03448042). 

### 12.7. Anti-HER2 Cancer Vaccines

The purpose of cancer vaccines is to activate the patient’s own immune system to identify and kill cancer cells. This is done by stimulating CD8+ and CD4+ T cells to respond to tumor-specific antigens, such as HER2. Cancer vaccines can range from simple peptides to complex autologous or allogenic cell-based vaccines. Several ongoing clinical trials are evaluating the safety and effectiveness of HER2-specific cancer vaccines, a few of which are detailed below. 

Nelipepimut-S is an MHC class I vaccine that consists of E75, a peptide derived from the extracellular domain of HER2, combined with granulocyte–macrophage colony stimulating factor (GM-CSF) as an immunoadjuvant. A phase I/II clinical trial evaluated the nelipepimut-S vaccine in patients with early-stage HER2-expressing breast cancer to prevent disease recurrence. In this trial, 195 patients with node-positive or high-risk node-negative HER2-expressing (IHC 1+, 2+, 3+) early-stage breast cancer, who completed all standard first-line therapies, received vaccination with nelipepimut-S. The vaccine was found to be well tolerated, with the majority of patients experiencing only grade 1 AEs [199]. Overall, there was only a numerical improvement in five-year DFS that was not statistically significant, with 89.7% in the vaccinated patients versus 80.2% for the unvaccinated controls (*p* = 0.08). However, 5-year DFS for optimally dosed vaccinated patients was 94.6% versus 87.1% for the unvaccinated controls (*p* = 0.05). Based on these positive outcomes, the phase III PRESENT trial was conducted to evaluate nelipepimut-S in patients with node-positive breast cancer and low-to-intermediate HER2 expression (IHC 1+/2+). However, no difference was observed in DFS in these HER2-low patients and the trial was terminated early [200].

WOKVAC is a plasmid-based DNA vaccine encoding epitopes from three breast cancer antigens: HER2, IGFBP2, and IGF1R. It is currently being investigated in phase II trials, including a study evaluating vaccine + paclitaxel + trastuzumab + pertuzumab as neoadjuvant therapy for HER2-positive breast cancer (NCT04329065). 

VRP-HER2 is a vaccine composed of an alphavirus vector encoding the ECD and transmembrane domains of HER2. In a preclinical xenograft translational study, VRP-HER2 demonstrated antitumor effects and increased HER2-specific memory CD8 T cells [201]. An ongoing phase II study is evaluating the vaccine in combination with pembrolizumab for patients with breast cancer (NCT03632941). 

Although HER2-specific cancer vaccines have not yet demonstrated efficacy in large trials, development for these vaccines is ongoing, and research continues to refine their design in the hopes of improving their outcomes. 

## 13. Conclusions

HER2 plays a pivotal role in the cell growth and proliferation of many cancer types, aside from breast cancer. The importance of knowing about HER2 is exemplified by the groundbreaking advancements that have been made, and the change in treatment plans it has brought to the oncological world in the last twenty years. Since its ground breaking discovery, there have been significant breakthroughs in knowledge regarding the actual receptor, the receptor’s biology, its mechanism of action, and advancements in tests to detect HER2, and significant strides in how to best incorporate targeted treatment. There are many ongoing studies currently happening which can continue to change the landscape of treatment of cancers effected by HER2 overexpression/proliferation. It is important to grasp the vast background and history of HER2, as well as the ongoing research in HER2, as this comprehension is an essential part of treating cancer today and comprehending the direction of future studies. 

## Figures and Tables

**Figure 1 ijms-25-01064-f001:**
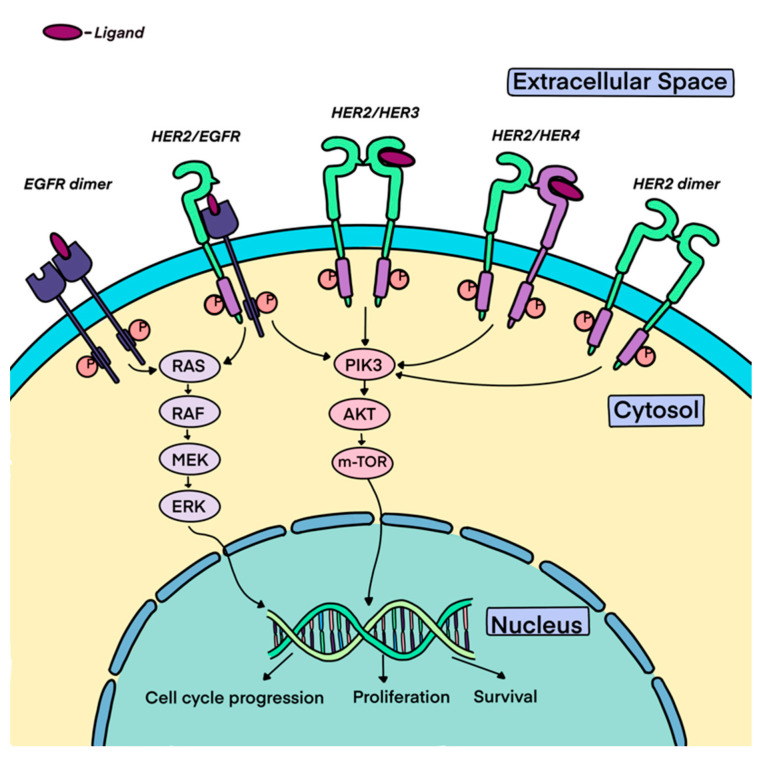
Downstream signaling pathway of the HER2 receptor.

**Figure 2 ijms-25-01064-f002:**
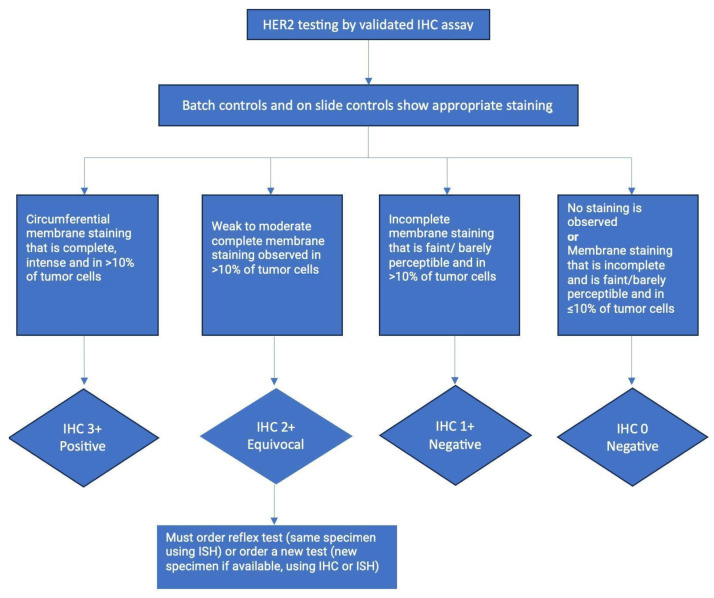
HER2 testing using IHC technique in breast cancer. Adapted from ASCO and CAP guidelines [21].

**Figure 3 ijms-25-01064-f003:**
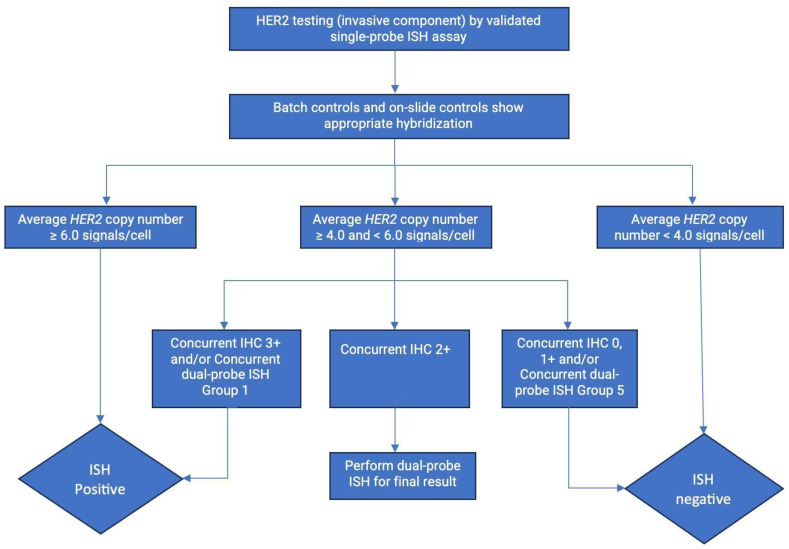
Single-probe ISH testing for evaluation of HER2 gene amplification in breast cancer. Adapted from ASCO and CAP guidelines [21]. (For groups, see Figure 4).

**Figure 4 ijms-25-01064-f004:**
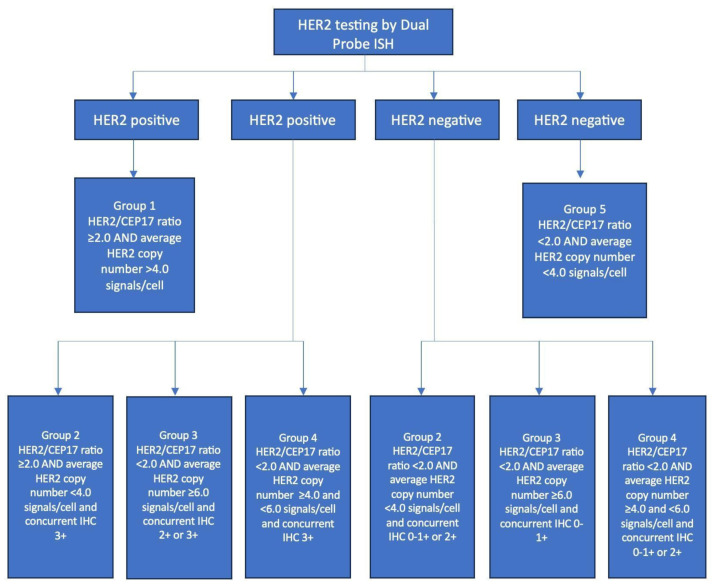
Dual-probe ISH testing for evaluation of HER2 gene amplification in invasive component of breast cancer specimen. Adapted and modified from ASCO and CAP guidelines [21]. CEP—Chromosome enumeration probe-targeting centromere in chromosome 17.

**Table 1 ijms-25-01064-t001:** Summary of selected trials of HER2-targeted therapy in cancers other than breast. NA = Not approved.

Study	Phase	Study Population	Number of Patients (Subgroup)	Intervention/Subgroup	PFS (Months)	OS (Months)	ORR (%)	DOR (Months)	FDA-Approval
Bang et al. TOGA trial 2010 [65]	III	Locally advanced/metastatic HER2-positive gastric or GE junction cancer, as first-line treatment	294	Trastuzumab + chemotherapy	6.7	13.8	47	6.9	20 October 2010
			290	Chemotherapy (cisplatin + 5FU or capecitabine)	5.5	11.1	35	4.8	
Thuss-Patience et al. GATSBY trial 2017 [66]	III	Locally advanced/metastatic HER2-positive gastric or GE junction cancer, progressed on first line	153	T-DM1	2.7	7.9	20.6	4.3	NA
			80	Taxanes	2.9	8.6	19.6	3.7	
Satoh et al. TyTAN trial 2014 [67]	III	Metastatic HER2-positive gastric cancer, progressed on first-line treatment	132	Lapatinib	5.5	11.0	27	7.4	NA
			129	Paclitaxel	4.4	8.9	9	5.1	
Hecht et al. TRIO-013/LOGiC 2017 [68]	III	Locally advanced or metastatic HER2-positive gastric or GE adenocarcinomas, as first line	249	Lapatinib + chemotherapy	6	12.2	53	7.3	NA
			238	Chemotherapy (capecitabine and oxaliplatin)	5.4	10.5	39	5.6	
Tabernero et al. JACOB trial 2018 [69]	III	Metastatic HER2-positive gastric or GEJ cancer, as first-line treatment	388	Pertuzumab + trastuzumab + chemotherapy	8.5	17.5	56.7	10.2	NA
			392	Trastuzumab + chemotherapy	7.0	14.2	48.3	8.4	
Shitara et al. DESTINY Gastric 01 2020 [70]	III	Advanced HER2-positive gastric or GE junction cancer, progressed on two prior lines including trastuzumab	126	T-DXd	5.6	12.5	51	11.3	15 January 2021
			62	Chemotherapy (paclitaxel or irinotecan)	3.5	8.4	14	3.9	
Janjigian et al. KEYNOTE 811 2021 [71]	III	Metastatic HER2-positive gastric or GE junction cancers, as first-line treatment	217	Pembrolizumab + trastuzumab + chemotherapy			74.4	10.6	5 May 2021
			216	Trastuzumab + chemotherapy (5FU + cisplatin or capecitabine + oxaliplatin)			51.9	9.5	
Sartore-Bianchi at al. HERACLES 2016 [72]	II	HER2-positive KRAS wild-type metastatic CRC, previously treated	27	Lapatinib + trastuzumab	21w	46w	30	38w	NA
Meric-Bernstam et al. MyPathwayCRC 2019 [73]	II	Metastatic HER2-positive CRC, previously treated	57	Trastuzumab and pertuzumab	2.9	11.5	32	5.9	NA
Sartore-Bianchi et al. HERACLES-B 2020 [74]	II	RAS/BRAF wild-type HER2-positive metastatic CRC, previously treated	31	Pertuzumab + TDM1	4.9		9.7		NA
Gupta et al. TAPUR 2022 [75]	II	HER2-amplified metastatic CRC, previously treated	28	Trastuzumab and pertuzumab	17.2 w	60 w	25		NA
Yoshino et al. DESTINY-CRC01 2022 [76]	II	HER2-positive RAS wild-type metastatic CRC, previously treated	53 Cohort A	T-DXd	6.9	15.5	45.3	7	NA
Strickler et al. MOUNTAINEER trial 2023 [77]	II	HER2-positive RAS wild-type unresectable or metastatic CRC, previously treated	84	Tucatinib + trastuzumab	8.2	24.1	38.1	12.4	19 January 2023
			31	Tucatinib			3.3		
Bob et al. DESTINY-Lung01 2022 [78]	II	HER2-mutant NSCLC refractory to standard treatment	91	T-DXd	8.2	17.8	55	9.3	11 August 2022
Bookman et al. 2003 [79]	II	HER2-positive epithelial ovarian or primary peritoneal carcinoma, previously treated	41	Trastuzumab	2		7.3	8w	NA
Fader et al. 2018/2020 [80,81]	II	HER2-positive advanced or recurrent endometrial cancer, either as primary treatment or for recurrent disease	30	Trastuzumab + carboplatin and paclitaxel	12.9	29.6			NA
			28	Carboplatin and paclitaxel	8	24.4			
Hussein et al. 2007 [82]	II	HER2-positive metastatic urothelial cancer as first-line treatment	44	Trastuzumab, paclitaxel, carboplatin, and gemcitabine	9.3	14.1	70	7.1	NA
Kurzrock et al. MyPathway 2020 [83]	II	Cohort of HER2-positive advanced salivary gland carcinoma, prior 0–3 lines	15	Trastuzumab + pertuzumab	8.6	20.4	60	9.2	NA
Javle et al. MyPathway 2021 [84]	II	Cohort of HER2-positive, biliary tract cancer, previously treated	39	Trastuzumab + pertuzumab	4	10.9	23	10.8	NA

**Table 2 ijms-25-01064-t002:** Level of evidence for ERBB2 alteration targetability according to OncoKB.

Level	HER2 Alterations	Cancer Types	Drugs
1	ERBB2 amplification	Breast cancer	Ado-Trastuzumab Emtansine
1	ERBB2 amplification	Breast cancer	Lapatinib + Capecitabine, Lapatinib + Letrozole
1	ERBB2 amplification	Breast cancer	Margetuximab + Chemotherapy
1	ERBB2 amplification	Breast cancer	Trastuzumab + Pertuzumab + Chemotherapy
1	ERBB2 amplification	Breast cancer	Trastuzumab + Tucatinib + Capecitabine
1	ERBB2 amplification	Breast cancer	Neratinib, Neratinib + Capecitabine
1	ERBB2 amplification	Breast cancer	Trastuzumab Deruxtecan
1	ERBB2 amplification	Breast cancer	Trastuzumab, Trastuzumab + Chemotherapy
1	ERBB2 amplification	Colorectal Cancer	Tucatinib + Trastuzumab
1	ERBB2 amplification	Esophagogastric Cancer	Pembrolizumab + Trastuzumab + Chemotherapy
1	ERBB2 amplification	Esophagogastric Cancer	Trastuzumab + Chemotherapy
1	ERBB2 amplification	Esophagogastric Cancer	Trastuzumab Deruxtecan
1	ERBB2 oncogenic mutations	Non-Small-Cell Lung Cancer	Trastuzumab Deruxtecan
2	ERBB2 amplification	Biliary tract cancer, NOS	Trastuzumab + Pertuzumab
2	ERBB2 amplification	Colorectal cancer	Lapatinib + Trastuzumab
2	ERBB2 amplification	Colorectal cancer	Trastuzumab + Pertuzumab
2	ERBB2 amplification	Colorectal cancer	Trastuzumab Deruxtecan
2	ERBB2 amplification	Uterine Serous Carcinoma/Uterine Papillary Serous Carcinoma	Trastuzumab + Carboplatin–Taxol Regimen
2	ERBB2 oncogenic mutations	Non-Small-Cell Lung Cancer	Ado-Trastuzumab Emtansine
3	ERBB2 oncogenic mutations	Breast cancer	Neratinib
3	ERBB2 oncogenic mutations	Non-Small-Cell Lung Cancer	Neratinib
3	ERBB2 oncogenic mutations	Non-Small-Cell Lung Cancer	Trastuzumab + Pertuzumab + Docetaxel

Level 1: FDA-recognized biomarker predictive of response to an FDA-approved drug in this indication; level 2: standard-of-care biomarker recommended by the NCCN or other professional guidelines predictive of the response to an FDA-approved drug in this indication; level 3: compelling clinical evidence supports the biomarker as being predictive of the response to a drug in this indication.

**Table 3 ijms-25-01064-t003:** Summary of FDA-approved anti-HER2 targeted agents.

Agent	MOA	U.S. Food and Drug Administration Indication	
		Type of Cancer	Setting/Line of Treatment (Year of Approval)
Trastuzumab	mAb	Breast cancer	Adjuvant (2006) or metastatic/1L (1998)
		Gastric cancer	Metastatic/1L (2010)
		Colorectal cancer *	Metastatic/2L (2023)
Pertuzumab	mAb	Breast cancer	Neoadjuvant, adjuvant or metastatic/1L (2012)
Margetuximab	mAb	Breast cancer	Metastatic/3L or later (2020)
Ado-trastuzumab emtansine (T-DM1)	ADC	Breast cancer	Adjuvant (2019) or metastatic/2L (2013)
Fam-trastuzumab deruxtecan-nxki (T-DXd)	ADC	Breast cancer	Unresectable or metastatic or neoadjuvant/adjuvant **/2L (2019) Unresectable or metastatic, HER2-low (2022)
		Gastric cancer	Locally advanced or metastatic/2L (2021)
		Non-small-cell lung cancer	Unresectable or metastatic, HER2-mutant (2022)
Tucatinib	TKI	Breast cancer	Advanced unresectable or metastatic (2020)
		Colorectal cancer	Unresectable or metastatic, RAS wild-type (2023)
Neratinib	TKI	Breast cancer	Extended adjuvant treatment (2017) Advanced or metastatic/3L or later (2020)
Lapatinib	TKI	Breast cancer	Advanced or metastatic/2L (2007)

MOA = mechanism of action; mAb = monoclonal antibody; ADC = antibody–drug conjugate; TKI = tyrosine kinase inhibitor; 1L = first line; 2L = second line; 3L = third line. * In combination with tucatinib. ** for patients who developed disease recurrence during or within 6 months of completing therapy.

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
