# Peer review of "Molecular Targeting of the Human Epidermal Growth Factor Receptor-2 (HER2) Genes across Various Cancers"

_ijms, 2024, doi:10.3390/ijms25021064_

Round 1

Reviewer 1 Report

Comments and Suggestions for Authors

In the review by Elizabeth Rubin et al. entitled “Molecular Targeting the Human Epidermal Growth Factor Receptor-2 (HER2) Genes Across Various Cancer”; the authors provide a comprehensive overview of the role of Human Epidermal Growth Factor Receptor 2 (HER2) in cancer, particularly its implication in various malignancies and its significance as a therapeutic target. The introduction effectively sets the stage by contextualizing HER2 within the ErbB family and highlighting its role in cellular signal transmission. The review demonstrates a commendable synthesis of information from over 200 published sources, contributing to the depth and credibility of the content. The discussion on the advancements in anti-HER2 therapy, particularly in breast cancer, is well-articulated, providing a clear understanding of the evolution of treatment strategies over the last two decades.

In general, the manuscript clearly outlines the current knowledge on the topic. Based on my assessment, with further adjustments, the manuscript could be considered for publication.

Below are some suggestions and corrections to improve the manuscript:

1. While reviewing the manuscript, I observed minor grammatical errors and typos that may impact overall readability. I recommend prioritizing improvements in grammar, punctuation, sentence structure, and spelling for a polished and error-free final version.

2. The manuscript demonstrates strong content, but I observed an inconsistency in the referencing format. It's advisable to place reference numbers before the sentence punctuation (period or comma) rather than after for a more standardized and reader-friendly citation style.

3. While Section 3, which discusses HER2 Biology, provides valuable insights, it would greatly enhance the credibility and depth of the content to include specific references supporting the information presented.

3. In the 'FDA Approval' column of Table 1, please use 'NA' or 'Not Approved' to indicate the absence of approval.

4. In addressing Section 11 on the Mechanism and Management of the most relevant toxicities, I recommend to enhance clarity and organization by subdividing it into multiple subsections with appropriate subtitles for better comprehension.

5. As for Section 12.2, 12.3, it is suggested to include references to support and validate the information provided.

6. The manuscript makes appropriate use of abbreviations, but it would be advisable to consider providing the full name or term when an abbreviation is used only once.

Comments on the Quality of English Language

Minor English language editing. I have noticed only some minor grammatical errors and typos.

Author Response

  1. While reviewing the manuscript, I observed minor grammatical errors and typos that may impact overall readability. I recommend prioritizing improvements in grammar, punctuation, sentence structure, and spelling for a polished and error-free final version: completed to the best of my ability.
  2. The manuscript demonstrates strong content, but I observed an inconsistency in the referencing format. It's advisable to place reference numbers before the sentence punctuation (period or comma) rather than after for a more standardized and reader-friendly citation style: completed 
  3. While Section 3, which discusses HER2 Biology, provides valuable insights, it would greatly enhance the credibility and depth of the content to include specific references supporting the information presented: references added 
  4. In the 'FDA Approval' column of Table 1, please use 'NA' or 'Not Approved' to indicate the absence of approval: completed
  5. In addressing Section 11 on the Mechanism and Management of the most relevant toxicities, I recommend to enhance clarity and organization by subdividing it into multiple subsections with appropriate subtitles for better comprehension: completed 
  6. As for Section 12.2, 12.3, it is suggested to include references to support and validate the information provided: completed/references added 

Reviewer 2 Report

Comments and Suggestions for Authors

The presented manuscript contains many positive aspects, including the topicality of the topic, which is targeted therapy. In the manuscript, the authors described in a very broad scope and with great precision the therapeutic progress in anti-HER2 therapy that has occurred not only in the case of breast cancer, but also in the case of other types of cancer. A positive aspect is also the presentation of the current direction for future research. The manuscript contains a large amount of detailed information that has been properly arranged to create a compact structure of a review article.

Below are comments regarding the reviewed manuscript:

1. The introduction failed to emphasize the purpose of the presented manuscript and its importance.

2. I propose to bold the names of individual drugs described in therapy to make them more readable.

You may also consider supplementing the manuscript with a graphical representation of the mechanisms of action of some drugs and showing their structure, which will make the manuscript even more attractive.

3. The manuscript should include:

-Author Contributions

-Funding

-Acknowledgments

-Conflicts of Interest

4. The last part of the manuscript lacked a very important element, which was a summary of the discussed problem.

In summary, the manuscript requires modifications before publication of the manuscript in IJMS.

Author Response

  • Introduction and conclusion were added to the paper.
  • The drugs names were not bolded as I heard mixed opinions on this. Since, the drugs are used so frequently in the manuscript there would be a very large amount of words bolded. From my understanding the editors also do not like when things are bolded unless it is a subsection. I am happy to change to bold of the drug names if ultimately that is what is wished by the paper, and there is a final consensus on this.  
  • I added 

    -Author Contributions

    -Funding

    -Acknowledgments

    -Conflicts of Interest

Round 2

Reviewer 2 Report

Comments and Suggestions for Authors

The manuscript has been corrected on the points I raised. I recommend this paper for publication.